# Photochemically produced SO$_2$ in the atmosphere of WASP-39b

Shang-Min Tsai[1,2✉], Elspeth K. H. Lee[3], Diana Powell[4], Peter Gao[5], Xi Zhang[6], Julianne Moses[7], Eric Hébrard[8], Olivia Venot[9], Vivien Parmentier[10], Sean Jordan[11], Renyu Hu[12,13], Munazza K. Alam[5], Lili Alderson[14], Natalie M. Batalha[15], Jacob L. Bean[16], Björn Benneke[17], Carver J. Bierson[18], Ryan P. Brady[19], Ludmila Carone[20], Aarynn L. Carter[15], Katy L. Chubb[21], Julie Inglis[13,22], Jérémy Leconte[23], Michael Line[18], Mercedes López-Morales[4], Yamila Miguel[24,25], Karan Molaverdikhani[26,27], Zafar Rustamkulov[28], David K. Sing[22,28], Kevin B. Stevenson[29], Hannah R. Wakeford[14], Jeehyun Yang[12], Keshav Aggarwal[30], Robin Baeyens[31], Saugata Barat[31], Miguel de Val-Borro[32], Tansu Daylan[33], Jonathan J. Fortney[15], Kevin France[34], Jayesh M. Goyal[35], David Grant[14], James Kirk[4,36], Laura Kreidberg[37], Amy Louca[24], Sarah E. Moran[38], Sagnick Mukherjee[15], Evert Nasedkin[37], Kazumasa Ohno[15], Benjamin V. Rackham[39,40], Seth Redfield[41], Jake Taylor[1,17], Pascal Tremblin[42], Channon Visscher[7,43], Nicole L. Wallack[5,13], Luis Welbanks[18], Allison Youngblood[44], Eva-Maria Ahrer[45,46], Natasha E. Batalha[47], Patrick Behr[34], Zachory K. Berta-Thompson[48], Jasmina Blecic[49,50], S. L. Casewell[51], Ian J. M. Crossfield[52], Nicolas Crouzet[24], Patricio E. Cubillos[20,53], Leen Decin[54], Jean-Michel Désert[31], Adina D. Feinstein[16], Neale P. Gibson[55], Joseph Harrington[56], Kevin Heng[26,46], Thomas Henning[37], Eliza M.-R. Kempton[57], Jessica Krick[58], Pierre-Olivier Lagage[42], Monika Lendl[59], Joshua D. Lothringer[60], Megan Mansfield[61], N. J. Mayne[62], Thomas Mikal-Evans[37], Enric Palle[63], Everett Schlawin[61], Oliver Shorttle[11], Peter J. Wheatley[45,46] & Sergei N. Yurchenko[19]

Photochemistry is a fundamental process of planetary atmospheres that regulates the atmospheric composition and stability[1]. However, no unambiguous photochemical products have been detected in exoplanet atmospheres so far. Recent observations from the JWST Transiting Exoplanet Community Early Release Science Program[2,3] found a spectral absorption feature at 4.05 μm arising from sulfur dioxide (SO$_2$) in the atmosphere of WASP-39b. WASP-39b is a 1.27-Jupiter-radii, Saturn-mass (0.28 $M_J$) gas giant exoplanet orbiting a Sun-like star with an equilibrium temperature of around 1,100 K (ref. 4). The most plausible way of generating SO$_2$ in such an atmosphere is through photochemical processes[5,6]. Here we show that the SO$_2$ distribution computed by a suite of photochemical models robustly explains the 4.05-μm spectral feature identified by JWST transmission observations[7] with NIRSpec PRISM (2.7$\sigma$)[8] and G395H (4.5$\sigma$)[9]. SO$_2$ is produced by successive oxidation of sulfur radicals freed when hydrogen sulfide (H$_2$S) is destroyed. The sensitivity of the SO$_2$ feature to the enrichment of the atmosphere by heavy elements (metallicity) suggests that it can be used as a tracer of atmospheric properties, with WASP-39b exhibiting an inferred metallicity of about 10× solar. We further point out that SO$_2$ also shows observable features at ultraviolet and thermal infrared wavelengths not available from the existing observations.

JWST observed WASP-39b as part of its Transiting Community Early Release Science Program (ERS Program 1366), with the goal of explaining its atmospheric composition[2,3]. Data from the NIRSpec PRISM and G395H instrument modes showed a distinct absorption feature between 4.0 μm and 4.2 μm, peaking at around 4.05 μm, that atmospheric radiative–convective–thermochemical equilibrium models could not explain with metallicity and C/O values typically assumed of gas giant planets (1–100× solar and 0.3–0.9, respectively)[8,9]. After excluding instrument systematics and stellar variability, a thorough search for gases has shown SO$_2$ as a promising candidate with the best-fit absorption feature (see Methods), although ad hoc spectra with injected SO$_2$ were used in the analysis.

Sulfur shares some chemical similarities with oxygen but uniquely forms various compounds with a wide range of oxidation states (−2 to +6 (ref. 10)). Although SO$_2$ is ubiquitously outgassed and associated with volcanism on terrestrial worlds (for example, Earth, Venus and Jupiter's satellite Io), the source of SO$_2$ is fundamentally different on gas giants. Under thermochemical equilibrium, sulfur chiefly exists

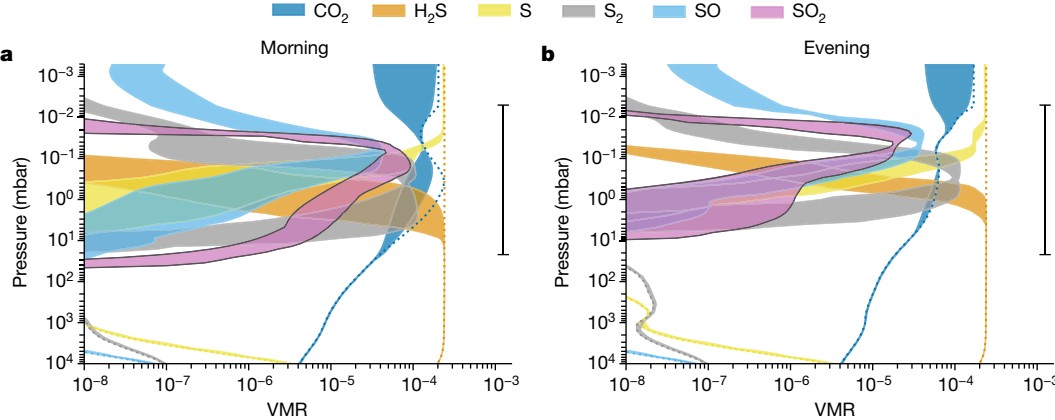

**Fig. 1 | Simulated vertical distribution of sulfur species and $CO_2$. a,b**, The colour-shaded areas indicate the span (enclosed by the maximum and minimum values) of VMRs of $CO_2$ (blue), $SO_2$ (pink with black borders) and other key sulfur species ($H_2S$, orange; S, yellow; $S_2$, grey; SO, light blue) computed by an ensemble of photochemical models (ARGO, ATMO, KINETICS and VULCAN) for the morning (**a**) and evening (**b**) terminators. The thermochemical equilibrium

VMRs are indicated by the dotted lines, with $SO_2$ not within the *x*-axis range owing to its very low abundance in thermochemical equilibrium. The range bar on the right represents the main pressure ranges of the atmosphere investigated by JWST NIRSpec spectroscopy. Photochemistry produces $SO_2$ and other sulfur species above the 1-mbar level with abundances several orders of magnitude greater than those predicted by thermochemical equilibrium.

in the reduced form, such that $H_2S$ is the primary sulfur reservoir in a hydrogen/helium-dominated gas giant[11–14]. At the temperature of WASP-39b, the equilibrium mixing ratio of $SO_2$ in the observable part of the atmosphere is less than about $10^{-12}$ for 10× solar metallicity and less than about $10^{-9}$ for even 100× solar metallicity (see Extended Data Fig. 1). This equilibrium abundance of $SO_2$ is several orders of magnitude smaller than the values needed to produce the spectral feature observed by JWST (volume mixing ratios (VMRs) of $10^{-6}$–$10^{-5}$)[8,9]. By contrast, under ultraviolet (UV) irradiation, $SO_2$ can be oxidized from $H_2S$ as a photochemical product. H and OH radicals, generated by photolysis processes, are key to liberating SH radicals and atomic S from $H_2S$ and subsequently oxidizing them to SO and $SO_2$. Although previous photochemical modelling studies have shown that substantial $SO_2$ can be produced in hydrogen-rich exoplanet atmospheres in this way[5,6,13,15,16], the extent to which such a model could reproduce the current WASP-39b observations remained unverified.

We have performed several independent, cloud-free 1D photochemical model calculations of WASP-39b using the ARGO, ATMO,

KINETICS and VULCAN codes (see Methods for model details). All models included sulfur kinetic chemical networks and were run using the same vertical temperature–pressure profiles of the morning and evening terminators adopted from a 3D WASP-39b atmospheric simulation with the Exo-FMS general circulation model (GCM)[17] (see Extended Data Fig. 2). The nominal models assumed a metallicity of 10× solar (ref. 18) with a solar C/O ratio of 0.55, whereas we explored the sensitivity to atmospheric properties.

The peak mixing ratios of the main sulfur species produced by the different photochemical models are largely consistent with each other to within an order of magnitude, as shown in Fig. 1. The $SO_2$ mixing ratio profiles are highly variable with altitude and strongly peaked at 0.01–1 mbar with a value of 10–100 ppm. $SO_2$ (along with $CO_2$) is more favoured at the cooler morning terminator, at which $H_2S$ is less stable against reaction with atomic H at depth (with $SO_2$ abundance peak of 50–90 ppm at the morning terminator and 15–30 ppm at the evening terminator). Although the peak $SO_2$ abundance from the photochemical models is greater than that estimated from fitting to the PRISM and

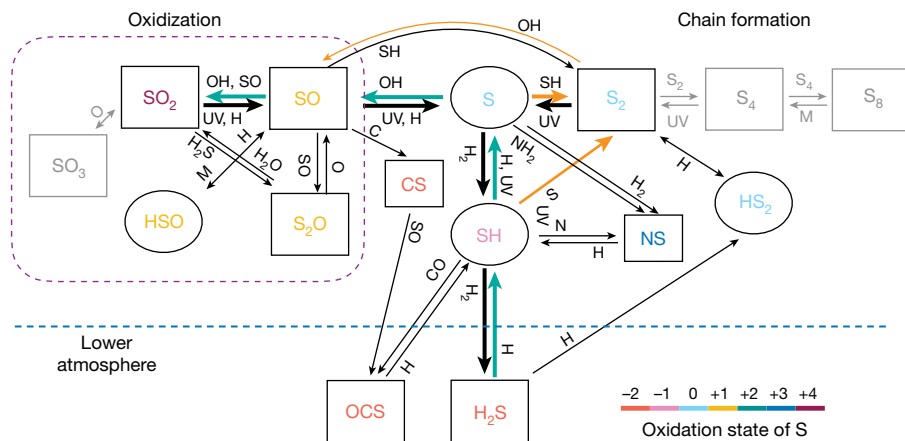

**Fig. 2 | A simplified schematic of the chemical pathways of sulfur species.** $H_2S$, which is the stable sulfur-bearing molecule at thermochemical equilibrium in an $H_2$ atmosphere, readily reacts with atomic H to form SH radicals and, subsequently, atomic S in the photochemical region (above about 0.1 mbar). Reaction of S with photochemically generated OH then produces SO, which is further oxidized to $SO_2$. The thick arrows denote efficient reactions and

M denotes any third body. Inefficient reactions and inactive paths in the temperature regime of WASP-39b are greyed out. The cyan arrows mark the main path from $H_2S$ to $SO_2$, whereas the orange arrows mark the paths that are important at higher pressures. Sulfur species are colour-coded by the oxidation states of S. Rectangles indicate stable molecules, whereas ovals indicate free radicals.

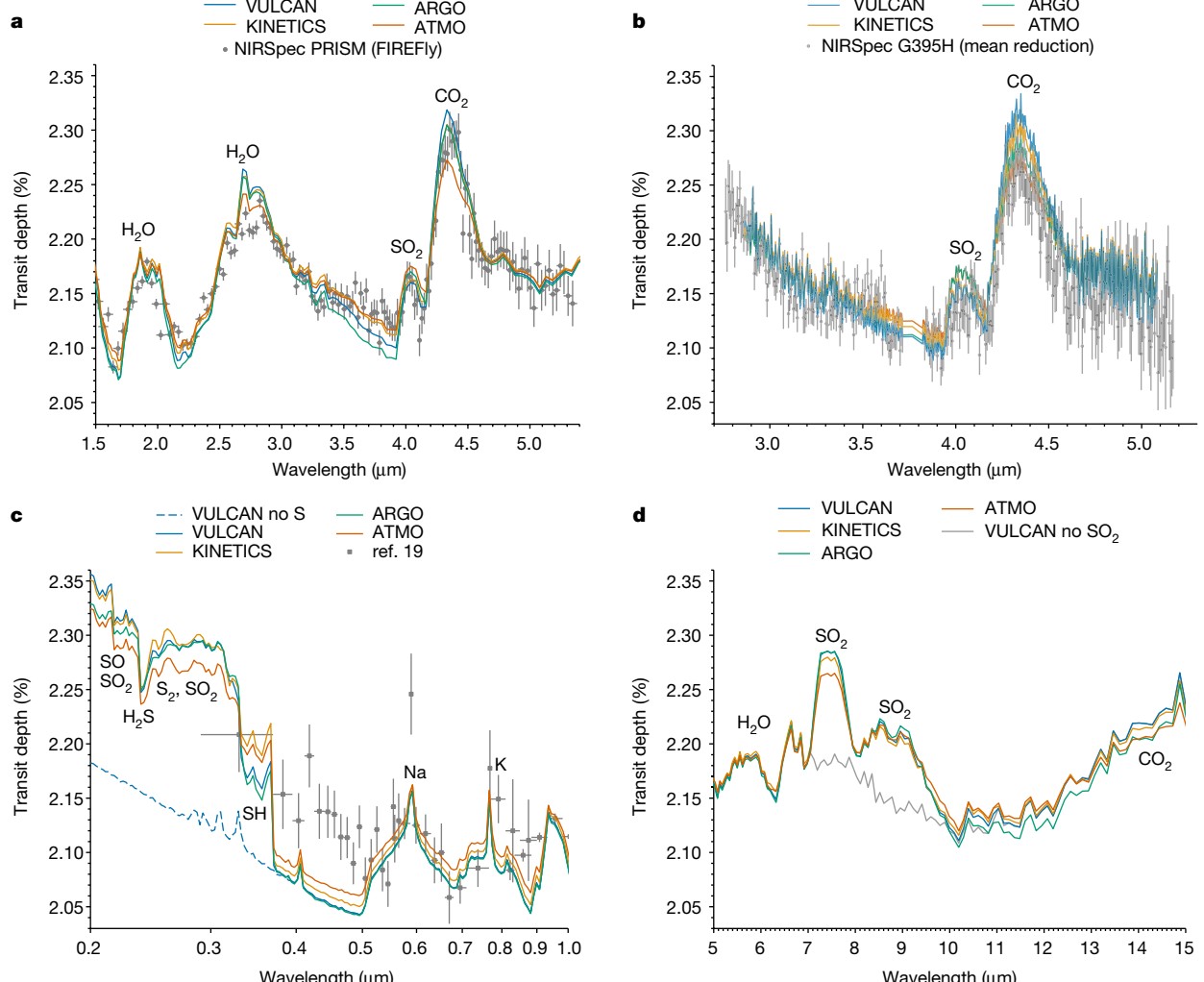

**Fig. 3 | Terminator-averaged theoretical transmission spectra.** We show the transmission spectra averaged over the morning and evening terminators generated from 1D photochemical model results. **a**, Comparison with the NIRSpec PRISM FIREFly reduction[8]. **b**, Comparison with the NIRSpec G395H weighted-mean reduction[9]. **c**, Comparison with the current HST and VLT/FORS2 optical wavelength data[19,37]. The models show pronounced features at UV wavelengths owing to sulfur species compared with the model without S-bearing species (dashed blue line). **d**, Predicted spectra across the MIRI LRS wavelength range, with $SO_2$ removed from the VULCAN output shown in grey to indicate its contribution. All of the spectral data show 1$\sigma$ error bars and the standard deviations averaged (unweighted) over all reductions are shown for the NIRSpec G395H data.

G395H data, which assumed vertically constant mixing ratios of about 1–10 ppm and about 2.5–4.6 ppm, respectively, the column-integrated number densities above 10 mbar are highly consistent (see Methods). Our models indicate that S, $S_2$ and SO, which are precursors of $SO_2$, also reach high abundances in the upper atmosphere above the pressure level at which $H_2S$ is destroyed. Nevertheless, they are not expected to manifest observable spectral features in the PRISM/G395H wavelength range.

The important pathways of sulfur kinetics in the atmosphere of WASP-39b from our models are summarized in Fig. 2. The photochemical production paths of $SO_2$ from $H_2S$ around the $SO_2$ peak are as follows:

$$
\begin{aligned}
H_2O &\xrightarrow{h\nu} OH + H \\
H_2O + H &\longrightarrow OH + H_2 \\
H_2S + H &\longrightarrow SH + H_2 \\
SH + H &\longrightarrow S + H_2 \\
S + OH &\longrightarrow SO + H \\
SO + OH &\longrightarrow SO_2 + H \\
\hline
\text{net: } H_2S + 2H_2O &\longrightarrow SO_2 + 3H_2
\end{aligned} \quad (1)
$$

Water photolysis in equation (1) is an important source of atomic H that initiates the pathway. The last step of oxidizing SO into $SO_2$ is generally the rate-limiting step. The oxidization of SO and photolysis of $SO_2$ account for the main sources and sinks of $SO_2$, which lead to altitude-varying distribution that peaks around 0.1 mbar (see Extended Data Fig. 4). At high pressures with less available OH, reactions involving $S_2$ become important in oxidizing S (see Methods). The growth of elemental sulfur allotropes beyond $S_2$ effectively stops for temperatures higher than approximately 750 K (refs. 5,6).

Figure 3 shows the morning/evening averaged transmission spectra resulting from the different photochemical models. All models are able to reproduce the strength and shape of the 4.05-µm $SO_2$ feature seen in the NIRSpec PRISM and G395H modes. The scatter in the model spectra is on par with the uncertainties of the data and is attributed to the spread in the vertical VMR structure of $SO_2$ and $CO_2$ produced by each model (Fig. 1). Also shown in Fig. 3 are the predicted spectra in the MIRI LRS wavelength range (5–12 µm), which exhibit prominent $SO_2$ features around 7.5 µm and 8.8 µm, as well as an upward slope redward of 12 µm owing to $CO_2$. Furthermore, our models predict a strong UV (0.2–0.38 µm) transmission signal from the presence of S species:

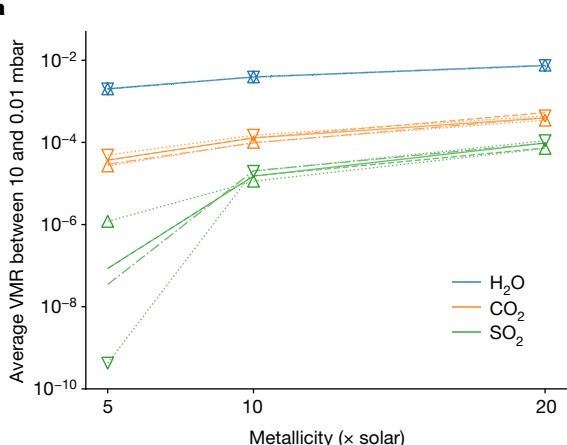

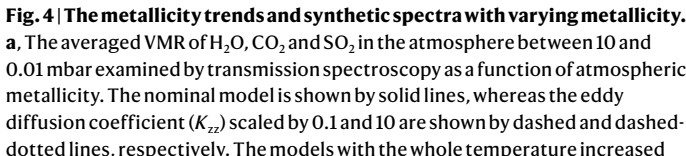

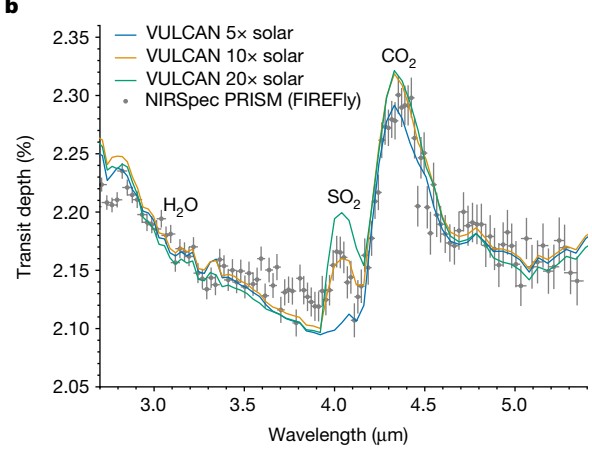

**Fig. 4 | The metallicity trends and synthetic spectra with varying metallicity.** **a**, The averaged VMR of $H_2O$, $CO_2$ and $SO_2$ in the atmosphere between 10 and 0.01 mbar examined by transmission spectroscopy as a function of atmospheric metallicity. The nominal model is shown by solid lines, whereas the eddy diffusion coefficient ($K_{zz}$) scaled by 0.1 and 10 are shown by dashed and dashed-dotted lines, respectively. The models with the whole temperature increased and decreased by 50 K are indicated by the upward-facing and downward-facing triangles connected by dotted lines, respectively. **b**, The morning and evening terminator-averaged theoretical transmission spectra with different metallicities (relative to solar value) compared with the NIRSpec observation. The error bars show $1\sigma$ standard deviations.

$H_2S$, $S_2$, $SO_2$ and SH produce a sharp opacity gradient shortward of 0.38 μm (Extended Data Fig. 7), at which the room-temperature UV cross-sections are used except those at 800 K for SH. The discrepancy between the models and previous HST STIS and VLT/FORS2 observations[19] (see Fig. 3) within 0.38–0.5 μm could potentially be because of enhanced UV opacities at high temperatures and/or aerosol particles. Further characterization of the sulfur species spectral features in the UV is promising with the scheduled HST/UVIS observation (Program 17162, principal investigators: Z. Rustamkulov and D. Sing).

$SO_2$ has recently been suggested as a promising tracer of metallicity in giant exoplanet atmospheres[16]. To test this and show trends in atmospheric properties, we have conducted sensitivity analysis on metallicity as well as temperature and vertical mixing using VULCAN (see Methods for details and further tests on C/O and stellar UV flux). Figure 4a summarizes these results for $SO_2$, along with $H_2O$ and $CO_2$, which are more commonly used as proxies for atmospheric metallicity[13,20–22]. Overall, the average abundance of $SO_2$ in the pressure region relevant for such observation is not strongly sensitive to temperature or vertical mixing once $SO_2$ has reached observable ppm levels and is mildly sensitive to C/O (see Extended Data Fig. 5). By contrast, $SO_2$ shows either a similar or a stronger dependence on metallicity compared with $H_2O$ and $CO_2$. This sensitivity to metallicity can be understood from the net reaction (equation (1)), in which it takes one molecule of $H_2S$ and two molecules of $H_2O$ to make one $SO_2$. Although $SO_2$ can be further oxidized into $SO_3$, which requires extra oxygen, $SO_3$ is rarely produced to an observable level in a $H_2$-dominated atmosphere. Therefore, $SO_2$ can be an ideal tracer of heavy-element enrichment for giant planets, with given constraints on the temperature and stellar far-ultraviolet (FUV) flux. The applicability of $SO_2$ as a tracer of metallicity is further shown in Fig. 4b, in which the increase in the $SO_2$ feature amplitude between 5× and 20× solar metallicity is much greater than that of $CO_2$ and $H_2O$. As such, retrieval analyses seeking to evaluate the atmospheric metallicity of warm giant exoplanets can substantially benefit from both $CO_2$ and $SO_2$ measurements.

Our results demonstrate the importance of considering photochemistry—and sulfur chemistry in particular—in warm exoplanet atmospheres when interpreting exoplanet atmospheric observations. Exoplanet photochemistry has been investigated using numerical models since the detection of an atmosphere on a transiting exoplanet[23,24], followed by a diverse set of subsequent studies explaining the interplay of carbon, oxygen, nitrogen, hydrogen and sulfur (see, for example,

ref. 25 for a review). It has been further pointed out that sulfur can affect other nonsulfur species, such as atomic H, $CH_4$ and $NH_3$ (refs. 6,15; also see Extended Data Fig. 6). Temperature trends in the photochemical production of sulfur species (Extended Data Fig. 10) in exoplanet atmospheres are potentially observable with features in the UV and infrared (Fig. 3 and Extended Data Fig. 7). At temperatures higher than that of WASP-39b, SH and SO may become relatively more abundant than $SO_2$ (refs. 6,13,15). Observing these compositional variations with temperature in $H_2$-dominated atmospheres, modulated by the atmospheric metallicity, could substantially improve our understanding of high-temperature chemical networks and atmospheric properties. The observational effort should also be complemented by a more accurate determination of key chemical reaction rate constants and UV cross-sections at the relevant temperatures (for example, refs. 26,27), as well as photochemical modelling development beyond 1D that includes horizontal transport (for example, refs. 28–30).

The accessibility of sulfur species in exoplanet atmospheres through the aid of photochemistry allows for a new window into planet-formation processes, whereas in the Solar System gas giants, the temperature is sufficiently low that sulfur is condensed out as either $H_2S$ clouds or together with $NH_3$ as ammonium hydrosulfide ($NH_4SH$) clouds[31], making it more difficult to observe. Sulfur has been detected in protoplanetary disks[32], in which it may be primarily in refractory form[33], making it a reference element showing the metallicity contributions of accreted solid and gas[34–36]. Such efforts for warm giant exoplanets are now a possibility thanks to the observability of photochemically produced $SO_2$. Thus, the detection of $SO_2$ offers valuable insights into further atmospheric characterization and planet formation.

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

[1]Atmospheric, Oceanic and Planetary Physics, Department of Physics, University of Oxford, Oxford, UK. [2]Department of Earth Sciences, University of California, Riverside, Riverside, CA, USA. [3]Center for Space and Habitability, University of Bern, Bern, Switzerland. [4]Center for Astrophysics | Harvard & Smithsonian, Cambridge, MA, USA. [5]Earth and Planets Laboratory, Carnegie Institution for Science, Washington, DC, USA. [6]Department of Earth and Planetary Sciences, University of California, Santa Cruz, Santa Cruz, CA, USA. [7]Space Science Institute, Boulder, CO, USA. [8]University of Exeter, Exeter, UK. [9]Université de Paris Cité and Univ. Paris Est Creteil, CNRS, LISA, Paris, France. [10]Université Côte d'Azur, Observatoire de la Côte d'Azur, CNRS, Laboratoire Lagrange, Nice, France. [11]Institute of Astronomy, University of Cambridge, Cambridge, UK. [12]Jet Propulsion Laboratory, California Institute of Technology, Pasadena, CA, USA. [13]Division of Geological and Planetary Sciences, California Institute of Technology, Pasadena, CA, USA. [14]School of Physics, University of Bristol, Bristol, UK. [15]Department of Astronomy and Astrophysics, University of California, Santa Cruz, Santa Cruz, CA, USA. [16]Department of Astronomy and Astrophysics, University of Chicago, Chicago, IL, USA. [17]Department of Physics and Institute for Research on Exoplanets, Université de Montréal, Montreal, Quebec, Canada. [18]School of Earth and Space Exploration, Arizona State University, Tempe, AZ, USA. [19]Department of Physics and Astronomy, University College London, London, UK. [20]Space Research Institute, Austrian Academy of Sciences, Graz, Austria. [21]Centre for Exoplanet Science, University of St Andrews, St Andrews, UK. [22]Department of Physics & Astronomy, Johns Hopkins University, Baltimore, MD, USA. [23]Laboratoire d'Astrophysique de Bordeaux, Université de Bordeaux, Pessac, France. [24]Leiden Observatory, University of Leiden, Leiden, the Netherlands. [25]SRON Netherlands Institute for Space Research, Leiden, the Netherlands. [26]Universitäts-Sternwarte München, Ludwig-Maximilians-Universität München, Munich, Germany. [27]Exzellenzcluster Origins, Munich, Germany. [28]Department of Earth & Planetary Sciences, Johns Hopkins University, Baltimore, MD, USA. [29]Johns Hopkins Applied Physics Laboratory, Laurel, MD, USA. [30]Indian Institute of Technology Indore, Indore, India. [31]Anton Pannekoek Institute for Astronomy, University of Amsterdam, Amsterdam, the Netherlands. [32]Planetary Science Institute, Tucson, AZ, USA. [33]Department of Astrophysical Sciences, Princeton University, Princeton, NJ, USA. [34]Laboratory for Atmospheric and Space Physics, University of Colorado Boulder, Boulder, CO, USA. [35]School of Earth and Planetary Sciences (SEPS), National Institute of Science Education and Research (NISER), Homi Bhabha National Institute (HBNI), Odisha, India. [36]Department of Physics, Imperial College London, London, UK. [37]Max Planck Institute for Astronomy, Heidelberg, Germany. [38]Lunar and Planetary Laboratory, University of Arizona, Tucson, AZ, USA. [39]Department of Earth, Atmospheric and Planetary Sciences, Massachusetts Institute of Technology, Cambridge, MA, USA. [40]Kavli Institute for Astrophysics and Space Research, Massachusetts Institute of Technology, Cambridge, MA, USA. [41]Astronomy Department and Van Vleck Observatory, Wesleyan University, Middletown, CT, USA. [42]Maison de la Simulation, CEA, CNRS, Univ. Paris-Sud, UVSQ, Université Paris-Saclay, Gif-sur-Yvette, France. [43]Chemistry and Planetary Sciences, Dordt University, Sioux Center, IA, USA. [44]NASA Goddard Space Flight Center, Greenbelt, MD, USA. [45]Centre for Exoplanets and Habitability, University of Warwick, Coventry, UK. [46]Department of Physics, University of Warwick, Coventry, UK. [47]NASA Ames Research Center, Moffett Field, CA, USA. [48]Department of Astrophysical and Planetary Sciences, University of Colorado Boulder, Boulder, CO, USA. [49]Department of Physics, New York University Abu Dhabi, Abu Dhabi, United Arab Emirates. [50]Center for Astro, Particle, and Planetary Physics (CAP3), New York University Abu Dhabi, Abu Dhabi, United Arab Emirates. [51]School of Physics and Astronomy, University of Leicester, Leicester, UK. [52]Department of Physics & Astronomy, University of Kansas, Lawrence, KS, USA. [53]INAF - Turin Astrophysical Observatory, Pino Torinese, Italy. [54]Institute of Astronomy, Department of Physics and Astronomy, KU Leuven, Leuven, Belgium. [55]School of Physics, Trinity College Dublin, Dublin, Ireland. [56]Planetary Sciences Group, Department of Physics and Florida Space Institute, University of Central Florida, Orlando, FL, USA. [57]Department of Astronomy, University of Maryland, College Park, MD, USA. [58]Infrared Processing and Analysis Center (IPAC), California Institute of Technology, Pasadena, CA, USA. [59]Département d'Astronomie, Université de Genève, Sauverny, Switzerland. [60]Department of Physics, Utah Valley University, Orem, UT, USA. [61]Steward Observatory, University of Arizona, Tucson, AZ, USA. [62]Department of Physics and Astronomy, Faculty of Environment, Science and Economy, University of Exeter, Exeter, UK. [63]Instituto de Astrofísica de Canarias (IAC), Tenerife, Spain. ✉e-mail: shangmit@ucr.edu

## Methods

### 4.05-μm feature

A list of gas species that have been compared with the 4.05-μm absorption feature in the transit observation of WASP-39b can be found in ref. 8. In particular, species with absorption features at similar wavelengths but are ruled out include $H_2S$, HCN, HBr, $PH_3$, SiO and $SiO_2$. $H_2S$ and HCN absorb shortward of the feature at 4.05 μm, whereas $SiO_2$ absorbs longward of that, and HBr, SiO and $PH_3$ have wider absorption bands than the observed feature. Chemically, SiO and $SiO_2$ are also expected to rain out at the temperature of WASP-39b and the solar elemental abundances have little bromine (Br/H $\approx 4 \times 10^{-10}$). Ultimately, the injection tests of $SO_2$ provide $2.7\sigma$ detection with NIRSpec PRISM (ref. 8) and $4.8\sigma$ with G395H (ref. 9).

### The temperature–pressure and eddy diffusion coefficient profiles derived from the Exo-FMS GCM

To provide inputs to the 1D photochemical models, a cloud-free WASP-39b GCM was run using the Exo-FMS GCM[17]. We computed the transmission spectra derived from our photochemical model results using gCMCRT (ref. 40) and the ExoAmes high-temperature $SO_2$ line list[41]. System parameters were taken from ref. 7. We assume a 10× solar metallicity atmosphere in thermochemical equilibrium and use two-stream, correlated-$k$ radiative transfer without optical and UV wavelength absorbers such as TiO, VO and Fe, which are assumed to have rained out from the atmosphere given the atmospheric temperatures of WASP-39b. The assumption about thermochemical equilibrium in radiative-transfer calculations will be discussed in the next section.

Although the temperatures of WASP-39b cross several condensation curves of sulfide clouds, such as $Na_2S$ and ZnS, the gas composition is not expected to be markedly affected. The elemental abundances of Na and Zn are less abundant than S (Na/S $\approx$ 0.13, Zn/S $\approx$ 0.0029), which would at most reduce approximately 20% of the total sulfur, similar to how oxygen is being sequestered in silicates and metals[42]. Furthermore, this full condensation is unlikely because sulfide condensates generally have high surface energies[43,44] that inhibit efficient nucleation, consistent with the detection of gaseous sodium on WASP-39b (ref. 8).

The radius of WASP-39b is inflated notably and we assume an internal temperature of 358 K, taken from the relationship between irradiated flux and internal temperature found in ref. 45. Extended Data Fig. 2a shows the latitude–longitude map of the temperature at a pressure level of 10 mbar. The input to the photochemical models are the temperature–pressure profiles at the morning and evening limbs (Extended Data Fig. 2), which we compute by taking the average of the profiles over all latitudes and ±10° (as estimated from the opening-angle calculations from ref. 46) of the morning (western) and evening (eastern) terminators (that is, the region between the grey curves in Extended Data Fig. 2a. The cooler morning terminator as a result of the horizontal heat transport facilitated by the global circulation can be seen in the figure.

Vertical mixing in 1D chemical models is commonly parameterized by eddy diffusion. For exoplanets, the eddy diffusion coefficient ($K_{zz}$) is in general a useful but loosely constrained parameter. For the 1D photochemical models used in this work, we assume that $K_{zz}$ follows an inverse square-root dependence with pressure in the stratosphere (for example, ref. 29) as

$$K_{zz}(\text{cm}^2\,\text{s}^{-1}) = 5 \times 10^7 \left( \frac{5\,\text{bar}}{P} \right)^{0.5} \quad (2)$$

and held constant below the 5-bar level in the convective zone. The eddy diffusion profile generally fits the root-mean-squared vertical wind multiplied by 0.1 scale height as the characteristic length scale from the GCM. The resulting $K_{zz}$ profile is presented in Extended Data Fig. 2.

### Radiative feedback of disequilibrium composition

The temperature profiles adopted from the GCM assume chemical equilibrium abundances. To evaluate the radiative feedback from disequilibrium chemical abundances, we first performed self-consistent 1D calculations, coupling the radiative-transfer and photochemical-kinetics models using HELIOS (ref. 47) and VULCAN (ref. 6), for which the opacity sources in HELIOS include $H_2O$, $CH_4$, CO, $CO_2$, $NH_3$, HCN, $C_2H_2$, SH, $H_2S$, $SO_2$, Na, K, $H^-$, CIA $H_2$–H2 and $H_2$–He (see references in ref. 47). Yet we found negligible differences between the temperature profile computed from equilibrium abundances and that from disequilibrium abundances. This is probably because water, as the predominant infrared opacity source, remains unaffected by disequilibrium processes. Meanwhile, a few opacities are missing in our radiative-transfer calculation. In particular, the opacity of $SO_2$ (ref. 48) does not extend into the visible and UV wavelength range. Previous works[13,49] indicated that SH and $S_2$ have strong absorption in the UV–visible and can potentially affect the thermal structure. To quantify the radiative effect of these sulfur species, we calculated the shortwave heating rate with

$$c_P \frac{dT}{dt} = \frac{F \kappa_i \Delta m_i}{\Delta m_{\text{air}}} \quad (3)$$

in which $c_P$ is the specific heat capacity of the air, $F$ is the stellar flux associated with the direct beam and $\Delta m_i$ and $\Delta m_{\text{air}}$ are the column mass of species $i$ and air of an atmospheric layer, respectively. Extended Data Fig. 3 illustrates the shortwave heating owing to SH, $S_2$ and $SO_2$. Our estimate shows that $SO_2$ contributed the most in our WASP-39b model, rather than SH and $S_2$ being the main shortwave absorbers for atmospheres with solar-like metallicity[13,49]. The peak of heating owing to $SO_2$ is comparable with a grey opacity of 0.05 $\text{cm}^2\,\text{g}^{-1}$ over 220–800 nm and could potentially raise the temperatures around 0.1 mbar (the visible grey opacity for the irradiation of WASP-39b irradiation is about 0.005 $\text{cm}^2\,\text{g}^{-1}$ (ref. 50)). Nevertheless, this heating effect does not change our main conclusions about photochemically forming $SO_2$ on WASP-39b. As long as temperatures do not fall below roughly 750 K, at which sulfur allotrope formation starts to take over, $SO_2$ is not too sensitive to temperature increases up to 100 K.

### The stellar spectrum of WASP-39

We require the high-energy spectral energy distribution (SED) of the WASP-39 host star as input to drive our set of photochemical models. However, as an inactive mid G-type star ($T_{\text{eff}}$ = 5,485 ± 50 K; ref. 51) at a distance of 215 pc (Gaia DR3), WASP-39 is too faint for high-S/N UV spectroscopy with HST. To approximate the stellar radiation incident on WASP-39b, we created a custom stellar SED that combines direct spectroscopy of WASP-39 in the optical (with HST/STIS G430L and G750L modes; GO 12473, principal investigator: D. Sing) with representative spectra from analogous stars at shorter wavelengths.

Our approach to estimating the UV stellar SED was based on two factors: (1) in the near-ultraviolet (NUV; 2,300–2,950 Å), in which the flux is dominated by the photosphere, we chose a proxy with a similar spectral type to WASP-39 and (2) in the extreme ultraviolet (XUV) and FUV (1–2,300 Å), in which the stellar flux is dominated by chromospheric, transition region and coronal emission lines, we chose a proxy star with similar chromospheric activity indicators and used spectral type as a secondary consideration. In the NUV, we used HST/STIS E230M spectra of HD 203244, a relatively active (Ca II $\log(R'_{HK})$ = −4.4 (ref. 52)), nearby (that is, unreddened, $d$ = 20.8 pc; Gaia DR2), G5 V star ($T_{\text{eff}}$ = 5,480 K (ref. 53)) from the STARCat archive[54]. Although HD 203244 is a suitable proxy at photospheric wavelengths, WASP-39 is a relatively old (about 7 Gyr) star with low chromospheric activity ($\log(R'_{HK})$ = −4.97 ± 0.06) and a long rotation period ($P_{\text{rot}}$ = 42.1 ± 2.6 days; ref. 51), suggesting substantially lower high-energy flux than HD

203244. Therefore, we elected to use a lower-activity G-type star, the Sun, at wavelengths shorter than 2,300 Å. The Sun has high-quality archival data available across the UV and X-rays and similar chromospheric activity to WASP-39 (the average solar Ca II $\log(R'_{HK})$ value is −4.902 ± 0.063 and ranges from approximately −4.8 to −5.0 from solar maximum to solar minimum[55,56]). With the components in hand, we first corrected the observed STIS spectra of WASP-39 for interstellar dust extinction of $E(B − V) = 0.079$ (ref. 57) using a standard $R_V = 3.1$ interstellar reddening curve[58] and then interpolated all spectra onto a 0.5-Å-pixel$^{-1}$ grid. The NUV spectrum of HD 203244 was scaled to the reddening-corrected WASP-39 observations in the overlap region between 2,900 and 3,000 Å and the XUV + FUV spectrum of the quiet Sun[59] was scaled to the blue end of the combined SED. The flux scaling between two spectral components is defined as $((F_{ref} − \alpha \times F_{proxy})/\sigma_{ref})^2$ in the overlap region, in which 'proxy' is the spectrum being scaled, 'ref' is the spectrum to which we are scaling and $\alpha$ is the scale factor applied to the proxy spectrum. $\alpha$ is varied until the above quantity is minimized ($\alpha = 2.04 \times 10^{-16}$ and $7.58 \times 10^{-3}$ for the FUV and NUV components, respectively). The final combined spectrum was convolved with a 2-Å full width at half maximum Gaussian kernel and wavelengths longer than 7,000 Å were removed to avoid the near-infrared fringing in the STIS G750L mode. We show the stellar spectrum at the surface of the star used for our photochemical models in Extended Data Fig. 2.

We compared our estimated SED for WASP-39 against archival GALEX observations from Shkolnik[60], who found the NUV (1,771–2,831 Å) flux density to be 168.89 µJy, or an average NUV spectral flux of $F_\lambda = 9.8 \times 10^{-16}$ erg cm$^{-2}$ s$^{-1}$ Å$^{-1}$ at 2,271 Å. Correcting this value by the average extinction correction in the GALEX NUV bandpass, a factor of 1.79, and comparing it with the average flux of our estimated SED over the same spectral range ($1.66 \times 10^{-15}$ erg cm$^{-2}$ s$^{-1}$ Å$^{-1}$), we find the agreement between the GALEX measurement of WASP-39 and our stellar proxy to be better than 6%.

### Simulated transmission spectra from gCMCRT
To post-process the 1D photochemical model output and produce transmission spectra, we use the 3D Monte Carlo radiative-transfer code gCMCRT[40].

For processing 1D columns, gCMCRT uses 3D spherical geometry but with a constant vertical profile across the globe in latitude and longitude. In this way, spectra from 1D outputs can be computed. We process the morning and evening terminator vertical 1D chemical profiles of each photochemical model separately, taking the average result of the two transmission spectra to produce the final spectra that are compared with the observational data.

In the transmission spectra model, we use opacities generated from the following line lists: $H_2O$ (ref. 61), OH (ref. 62), CO (ref. 63), $CO_2$ (ref. 64), $CH_4$ (ref. 65), $CH_3$ (ref. 66), HCN (ref. 67), $C_2H_2$ (ref. 68), $C_2H_4$ (ref. 69), $C_2H_6$ (ref. 70), $C_4H_2$ (ref. 70), $C_2$ (ref. 71), CN (ref. 72), CH (ref. 73), $SO_2$ (ref. 41), SH (ref. 48), SO (ref. 74), $H_2S$ (ref. 75), NO (ref. 76), $N_2O$ (ref. 76), $NO_2$ (ref. 76), HCl (ref. 70), Na (ref. 77), K (ref. 77).

### Description of photochemical models
We use the following 1D thermo-photochemical models to produce the steady-state chemical abundance profiles for the terminators of WASP-39b. All models assume cloud-free conditions and adopt the same temperature profiles, stellar UV flux, eddy diffusion coefficient profile (Extended Data Fig. 2) and zero-flux (closed) boundary conditions. A zenith angle of 83° (an effective zenith angle that matches the terminator-region-mean actinic flux for near-unity optical depth) is assumed for the terminator photochemical modelling.

**VULCAN.** The 1D kinetics model VULCAN treats thermochemical[78] and photochemical[6] reactions. VULCAN solves the Eulerian continuity equations, including chemical sources/sinks, diffusion and advection transport, and condensation. We applied the C-H-N-O-S network

(https://github.com/exoclime/VULCAN/blob/master/thermo/SNCHO_photo_network.txt) for reduced atmospheres containing 89 neutral C-bearing, H-bearing, O-bearing, N-bearing and S-bearing species and a total of 1,028 thermochemical reactions (that is, 514 forward–backward pairs) and 60 photolysis reactions. The sulfur allotropes are simplified into a system of S, $S_2$, $S_3$, $S_4$ and $S_8$. The sulfur kinetics data are drawn from the NIST and KIDA databases, as well as modelling[5,79] and ab initio calculations published in the literature (for example, ref. 80). For simplicity and cleaner model comparison, the temperature-dependent UV cross-sections[6] are not used in this work. The pathfinding algorithm described in ref. 81 is used to identify the important chemical pathways. We note that the paths presented in this study are mainly based on VULCAN output (see Extended Data Table 1). Although detailed reactions might differ between different photochemical models, the main paths remain robust.

**KINETICS.** The KINETICS 1D thermo-photochemical transport model[42] uses the Caltech/JPL KINETICS model[82,83] to solve the coupled 1D continuity equations describing the chemical production, loss and vertical transport of atmospheric constituents of WASP-39 b. The model contains 150 neutral C-bearing, H-bearing, O-bearing, N-bearing, S-bearing and Cl-bearing species that interact with each other through a total of 2,350 reactions (that is, 1,175 forward–reverse reaction pairs). These reactions have all been fully reversed through the thermodynamic principle of microscopic reversibility[84], such that the model would reproduce thermochemical equilibrium in the absence of transport and external energy sources, given sufficient integration time. The chemical reaction list involving C-bearing, H-bearing, O-bearing and N-bearing species is taken directly from ref. 22. Included for the first time here are 41 sulfur and chlorine species: S, S(1D), $S_2$, $S_3$, $S_4$, $S_8$, SH, $H_2S$, $HS_2$, $H_2S_2$, CS, $CS_2$, HCS, $H_2CS$, $CH_3S$, $CH_3SH$, SO, $SO_2$, $SO_3$, $S_2O$, $HOSO_2$, $H_2SO_4$ (gas and condensed), OCS, NS, NCS, HNCS, Cl, $Cl_2$, HCl, ClO, HOCl, ClCO, $ClCO_3$, ClS, $ClS_2$, $Cl_2S$, ClSH, OSCl, $ClSO_2$ and $SO_2Cl_2$. The thermodynamic data of several chlorine-bearing and sulfur-bearing species are not available in the previous literature and we performed ab initio calculations for these species. We first carried out electronic-structure calculations at the CBS-QB3 level of theory using Gaussian 09 (ref. 85) to determine geometric conformations, energies and vibrational frequencies of the target molecules. Then the thermodynamic properties of these molecules were calculated by Arkane (ref. 86), a package included in the open-source software RMG v3.1.0 (refs. 87,88), with atomic-energy corrections, bond corrections and spin–orbit corrections, based on the CBS-QB3 level of theory as the model chemistry. The reaction rate coefficients and photolysis cross-sections for these S and Cl species are derived from Venus studies[89–94], interstellar medium studies[95], Io photochemical models[96,97], Jupiter cometary-impact models[98,99], the combustion-chemistry literature[100–103], terrestrial stratospheric compilations[104,105] and numerous individual laboratory or computational kinetics studies (such as refs. 106–110).

**ARGO.** The 1D thermochemical and photochemical kinetics code ARGO originally[111,112] used the Stand2019 network for neutral hydrogen, carbon, nitrogen and oxygen chemistry. ARGO solves the coupled 1D continuity equation including thermochemical-photochemical reactions and vertical transport. The Stand2019 network was expanded by Rimmer et al.[113] by updating several reactions, incorporating the sulfur network developed by ref. 15, and supplementing it with reactions from refs. 93,114, to produce the Stand2020 network. The Stand2020 network includes 2,901 reversible reactions and 537 irreversible reactions, involving 480 species composed of H, C, N, O, S, Cl and other elements.

**ATMO.** The C-H-N-O chemical kinetics scheme from ref. 115 is implemented by ref. 116 in the standard 1D atmosphere model ATMO, which solves for the chemical disequilibrium steady state. As of the time of

writing of this article, the sulfur kinetic scheme of ATMO, derived from applied combustion models, is still at the development and validation stage. Hence, for WASP-39b, we performed ATMO with the C-H-N-O-S thermochemical network from VULCAN (ref. 6) along with the photochemical scheme from ref. 117 (an update of the native photochemical scheme from ref. 115), with another 71 photolysis reactions of $H_2S$, $S_2$, $S_2O$, SO, $SO_2$, $CH_3SH$, SH, $H_2SO$ and COS.

## Sensitivity tests

We examine the sensitivity of our chemical outcomes to essential atmospheric properties using VULCAN. For models with various metallicity and C/O ratios, we explore the sensitivity to temperature and vertical mixing by systematically varying the temperature–pressure and eddy diffusion coefficient profiles. Specifically, the temperature throughout the atmosphere is shifted by 50 K and the eddy diffusion coefficients are multiplied/divided by 10. These variations span a range comparable with the temperature differences among radiative transfer models[47] and the uncertainties in parameterizing vertical mixing with eddy diffusion coefficients[118,119]. On our choice of internal heat, we have further conducted tests with different internal temperatures and found that the compositions above 1 bar are not sensitive to internal temperature, because the quench levels of the main species are at higher levels given the adopted eddy diffusion coefficient. We have also verified that the temperature above the top boundary of the GCM (about $5 \times 10^{-5}$ bar; Extended Data Fig. 2) does not affect the composition below.

Sensitivity to C/O is summarized in Extended Data Fig. 5, in which the nominal model has a C/O ratio of 0.55, as in the main text. The averaged abundance of both $SO_2$ and $H_2O$ in the pressure region relevant for transmission spectrum observations show similar dependencies on C/O, decreasing by a few factors as the C/O increased from sub-solar (0.25) to super-solar (0.75) values. The averaged abundance of $SO_2$ is not very sensitive to temperature and vertical mixing either, except for C/O = 0.75, for which the $SO_2$ concentration is at roughly the ppm level, similar to what is found in Fig. 4.

Finally, we performed sensitivity tests to the UV irradiation—the ultimate energy source of photochemistry. We first tested the sensitivity to the assumed stellar spectra by performing the same models with the solar spectrum (close to WASP-39) and found negligible differences in the photochemical results. Because the UV spectrum shortward of 295 nm is constructed from stellar proxies rather than directly measured, we then focused on varying the stellar flux in the FUV (1–230 nm) and NUV (230–295 nm) separately. Extended Data Fig. 8 shows that the resulting sulfur species abundances are almost identical when the UV flux is reduced by a factor of 10, broadly consistent with what Zahnle et al.[5] suggested that the photochemical destruction of $H_2S$ only becomes photon-limited when the stellar UV flux is reduced by about two orders of magnitude (for a directly imaged gas giant). On the other hand, although SO and $SO_2$ are not sensitive to increased NUV, they are substantially depleted with increased FUV. This is because the photodissociation of SO and $SO_2$ mainly operates in the FUV and the enhanced FUV can destroy SO and $SO_2$, even with the same amount of available OH radicals.

## Spectral effects of assuming a vertically uniform $SO_2$ distribution

Minor species commonly have VMR varying with altitude in the observable region of the atmosphere, especially those produced or destroyed by photochemistry. Extended Data Fig. 9 demonstrates that assuming a vertically constant VMR of $SO_2$ can lead to underestimating its abundances by about an order of magnitude. This is verified by comparing the column-integrated number density from the pressure level relevant for transmission spectroscopy. For example, the terminator-averaged column-integrated number density of $SO_2$ above 10 mbar by VULCAN is about $1.4 \times 10^{19}$ molecules cm$^{-2}$, which is equal to a vertically uniform $SO_2$ with a concentration around 4 ppm. Hence modelling frameworks

that assume vertically uniform composition should be treated with caution and would benefit from comparisons with photochemical models, especially for photochemically active species that can exhibit large vertical gradients.

## Opacities of sulfur species

The opacities of sulfur species illustrated in Extended Data Fig. 7 are compiled from UV cross-sections and infrared line lists. The room-temperature UV cross-sections are taken from the Leiden Observatory database[120] (http://home.strw.leidenuniv.nl/~ewine/photo). The infrared opacities include $SO_2$ (ref. 121), $H_2S$ (refs. 122,48), CS (ref. 123) and a newly computed high-temperature line list for SO (ref. 74). The opacity from OCS (ref. 124) is only available up to room temperature at present, hence its coverage is probably incomplete in our region of interest.

## Alternative $SO_2$ production pathways

$S_2$ formation can compete with $SO_2$ production, as we will explore in detail in the next section. On WASP-39b, reactions involving $S_2$ are found to be important in oxidizing S at high pressures at which less OH is available. S and SH would first react to form $S_2$ by SH + S → H + $S_2$ before getting oxidized through $S_2$ + OH → SO + SH. The scheme is similar to that in equation (1) except SH and $S_2$ play the role of the catalyst to oxidize S into SO, whereas SO can also self-react to form $SO_2$ in this regime (references of important reactions are listed in Extended Data Table 1).

## Implications of observing sulfur photochemistry

The temperature of WASP-39b resides within the sweet spot of producing $SO_2$ (ref. 16). Previous photochemical modelling works suggested that, at lower temperatures, sulfur allotropes would be favoured over $SO_2$, whereas SH can prevail at higher temperatures[5,6]. Here we briefly explain the general temperature trends of sulfur photochemical products.

After S is liberated from $H_2S$, sulfur can follow either the oxidization or the chain polymerization paths, as illustrated in Fig. 2. The competing of the two paths is essentially controlled by the abundance of the oxidizing radical OH relative to atomic H. We can estimate the OH to H ratio by assuming that OH is in quasi-equilibrium with $H_2O$, that is, $k_{H_2O}[H_2O][H] = k'_{H_2O}[OH][H_2]$, in which $k_{H_2O}$ and $k'_{H_2O}$ are the forward and backward rate constants of $H_2O$ + H → OH + $H_2$, respectively. Then, $[OH]/[H] \approx 2\frac{k_{H_2O}}{k'_{H_2O}} \times O/H$, because most of the O is in $H_2O$. Extended Data Fig. 10a shows that the [OH]/[H] ratio strongly depends on temperature. When the temperature drops below about 750 K, the scarcity of OH makes S preferably react with SH to form $S_2$. SO and $SO_2$ could only be produced at higher altitudes, at which more OH is available from water photolysis (for example, refs. 5,6).

We further perform photochemical calculations using VULCAN with a grid of temperature profiles across planetary equilibrium temperatures 600–2,000 K, adopted from the 1D radiative–convective equilibrium models applied in ref. 39, in which an internal temperature of 100 K with perfect heat redistribution and gravity $g = 1,000$ cm s$^{-2}$ are assumed. Apart from the thermal profiles, we keep the rest of the planetary parameters the same as the WASP-39b model in this work, including stellar UV irradiation. Extended Data Fig. 10b reveals the observation of sulfur photochemistry on other irradiated exoplanets, summarizing the averaged abundances of the key sulfur molecules produced by photochemistry as a function of equilibrium temperature. For 10× solar metallicity, the sweet-spot temperature for producing observable $SO_2$ is 1,000 K $\lesssim T_{eq} \lesssim$ 1,600 K. For $T_{eq} \lesssim$ 1,000 K, $SO_2$ production below the 0.01-mbar level ceased and $S_x$ (sulfur allotropes; mainly $S_2$ and $S_8$ here) is more favoured. For $T_{eq} \gtrsim$ 1,600 K, SH becomes the predominant sulfur-bearing molecular (apart from atomic S) around mbar levels. Although observing SH is challenging in the infrared, it can potentially be identified in the near-UV (300–400 nm)[125].

## Data availability

The data used in this paper are associated with JWST ERS Program 1366 and are available from the Mikulski Archive for Space Telescopes (https://mast.stsci.edu), which is operated by the Association of Universities for Research in Astronomy, Inc., under NASA contract NAS 5-03127 for JWST. The chemical networks and abundance output of the photochemical models (ARGO, ATMO, KINETICS and VULCAN) presented in this study are available at https://doi.org/10.5281/zenodo.7542781.

## Code availability

The codes VULCAN and gCMCRT used in this work to simulate composition and produce synthetic spectra are publicly available: VULCAN[6,78] (https://github.com/exoclime/VULCAN); gCMCRT[40] (https://github.com/ELeeAstro/gCMCRT).

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

**Acknowledgements** This work is based on observations made with the NASA/ESA/CSA JWST. The working groups are associated with programme JWST-ERS-01366. The initial manuscript was improved by the constructive comments from L. Mancini, J. Mendonça, A. Saba and X. Tan. S.-M.T. is supported by the European Research Council advanced grant EXOCONDENSE (no. 740963; principal investigator: R. T. Pierrehumbert). E.K.H.L. is supported by the SNSF Ambizione Fellowship grant (no. 193448). X.Z. is supported by NASA Exoplanet Research grant 80NSSC22K0236. O.V. acknowledges funding from the ANR project 'EXACT' (ANR-21-CE49-0008-01), from the Centre National d'Études Spatiales (CNES) and from the CNRS/INSU Programme National de Planétologie (PNP). L.D. acknowledges support from the European Union H2020-MSCA-ITN-2109 under grant no. 860470 (CHAMELEON) and the KU Leuven IDN/19/028 grant Escher. This work benefited from the 2022 Exoplanet Summer Program at the Other Worlds Laboratory (OWL) at the University of California, Santa Cruz, a programme financed by the Heising-Simons Foundation. T.D. is an LSSTC Catalyst Fellow. J.K. is an Imperial College Research Fellow. B.V.R. is a 51 Pegasi b Fellow. L.W. is an NHFP Sagan Fellow. A.D.F. is an NSF Graduate Research Fellow.

**Author contributions** All authors played a notable role in the JWST Transiting Exoplanet Community Early Release Science Program, including the original proposal, preparatory work, tool development, coordinating meetings and so on. Some specific contributions are listed as follows. S.-M.T., P.G., D.P., X.Z., E.K.H.L. and V.P. designed the project and drafted the article. E.K.H.L. and L.C. performed 3D GCMs. S.-M.T., J.M., E.H., O.V., S.J., R.H., J.Y., K.M., R.B., C.J.B. and A.L. developed and/or performed photochemical models. S.-M.T., J.M., E.H., O.V., S.J., R.H., K.O. and P.T. contributed substantially to model comparisons and chemical analysis. K.L.C. and S.-M.T. compiled the sulfur opacities and E.K.H.L. computed the synthetic spectra. Z.R., D.K.S., J.K., E.S., and A.L.C. reduced and analysed the NIRSpec PRISM data. L.A., H.R.W., M.K.A., S.B., D.G., J.I., T.M.-E. and N.L.W. reduced and analysed the NIRSpec G395H data, with further contributions from J.B. and T.D. B.V.R., J.J.F., S.E.M., S.R., Y.M., K.L.C. and L.D. provided substantial feedback, with E.H. coordinating comments from all other authors, to improve the manuscript.

**Competing interests** The authors declare no competing interests.

**Additional information**
**Correspondence and requests for materials** should be addressed to Shang-Min Tsai.

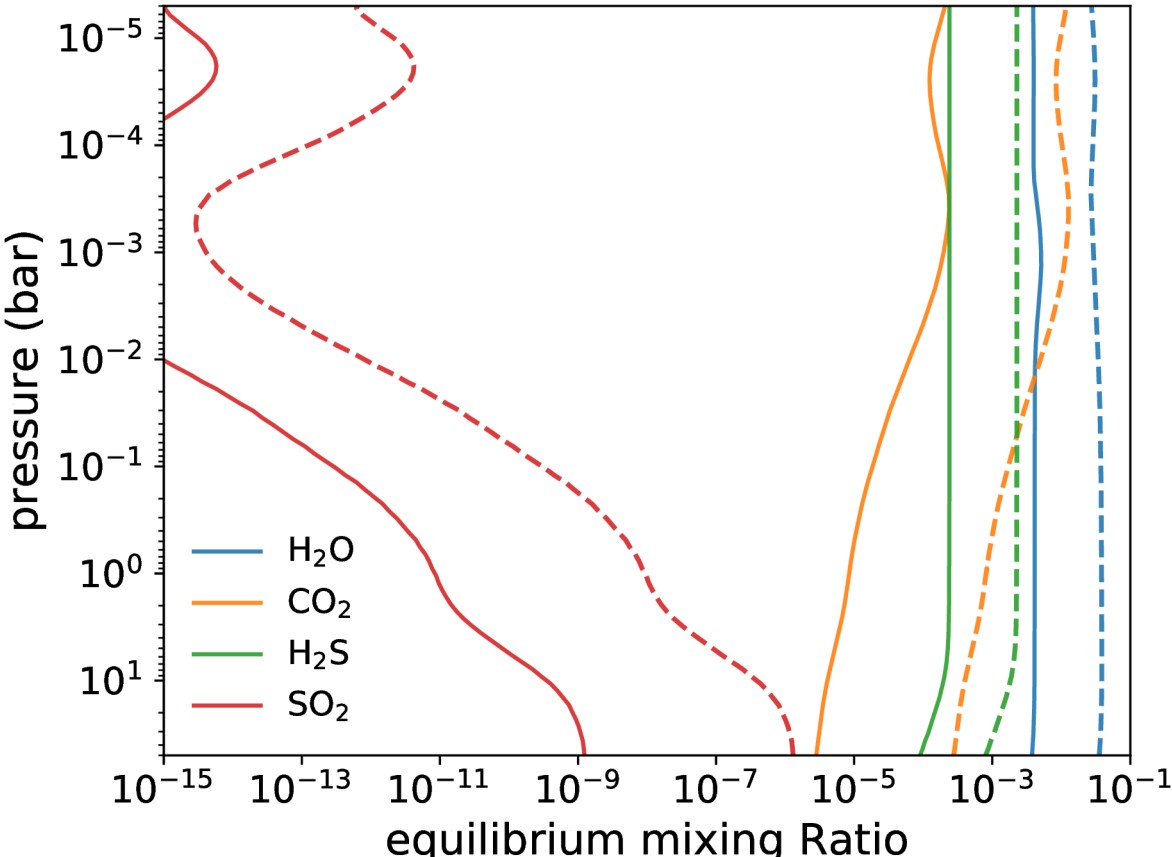

**Extended Data Fig. 1 | Chemical equilibrium abundances in the atmosphere of WASP-39b.** VMR profiles of $H_2O$ (blue), $CO_2$ (orange), $H_2S$ (green) and $SO_2$ (red), as computed by FastChem (ref. 38) based on the morning terminator temperature profile, are given for 10× (solid lines) and 100× (dashed lines) solar metallicity.

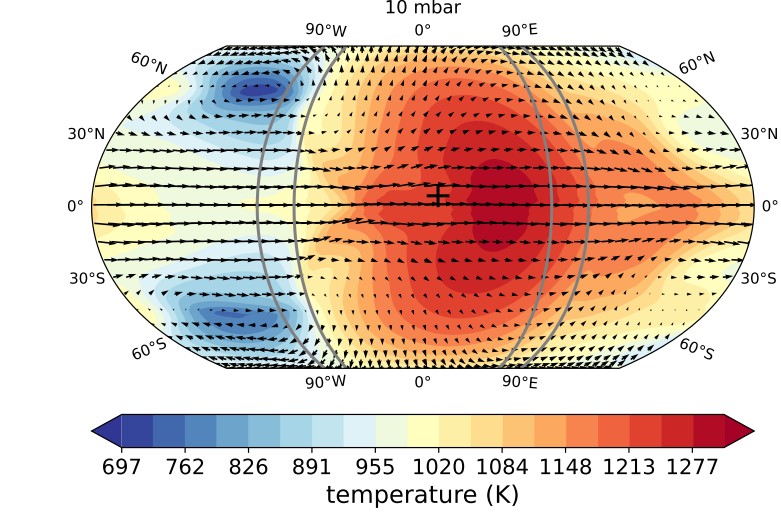

(a)

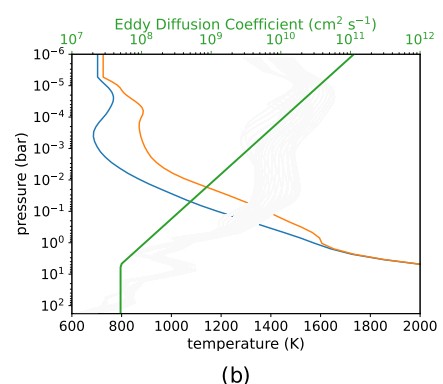

(b)

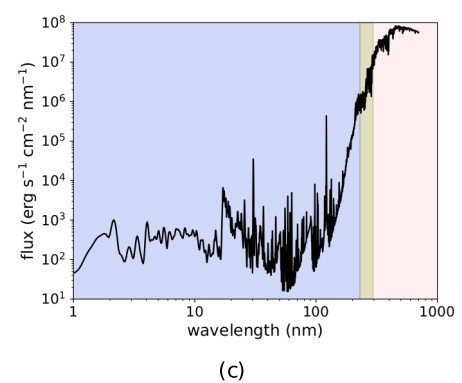

(c)

**Extended Data Fig. 2 | The temperature–wind map of the WASP-39b Exo-FMS GCM and input for 1D photochemical models. a**, The colour scale represents temperature across the planet and arrows denote the wind direction and magnitude at 10 mbar. The ±10° longitudinal regions with respect to the morning and evening terminators are indicated with solid grey lines. The '+' symbol denotes the sub-stellar point. **b**, 1D temperature–pressure profiles adopted from the morning and evening terminators averaging all latitudes and ±10° longitudes (regions enclosed by grey lines in **a**) and the $K_{zz}$ profile (equation (2) and held constant below the 5-bar level) overlaying the root-mean-squared vertical wind multiplied by 0.1 scale height from the GCM (grey). The temperatures are kept isothermal from those at the top boundary of the GCM around $5 \times 10^{-5}$ bar when extending to lower pressures (about $10^{-8}$ bar) for photochemical models. **c**, Input WASP-39 stellar flux at the surface of the star. The pink-shaded region indicates the optical wavelength range at which the stellar spectrum is directly measured, whereas the blue-shaded and green-shaded regions are those constructed from the Sun and HD 203244, respectively.

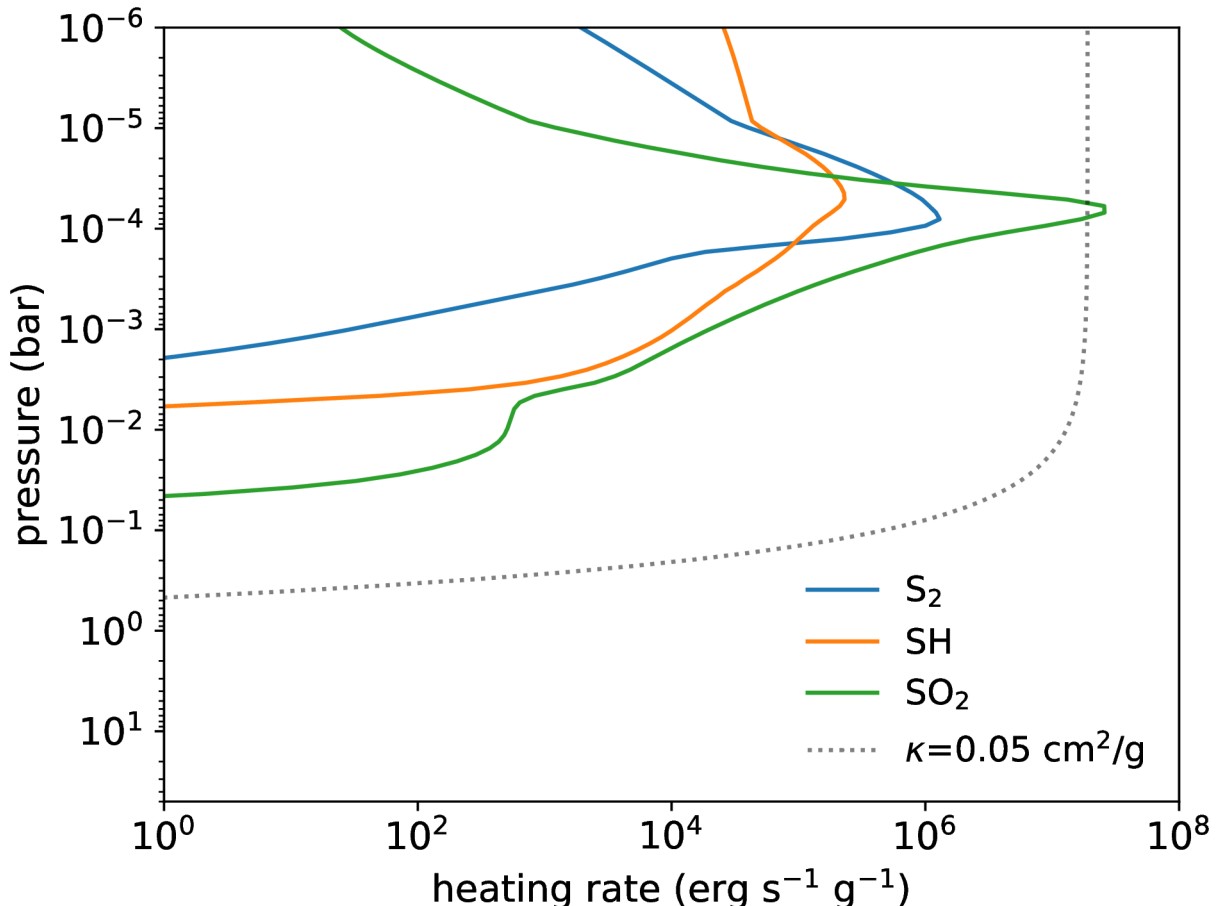

**Extended Data Fig. 3 | Shortwave radiative heating of sulfur species.** Radiative heating rates (erg $s^{-1}$ $g^{-1}$) of $SO_2$, SH and $H_2S$ to demonstrate their potential impact on the temperature structure. Heating owing to a vertically constant grey opacity of 0.05 $cm^2$ $g^{-1}$ is shown for comparison. All heating rates are integrated over 220–800 nm.

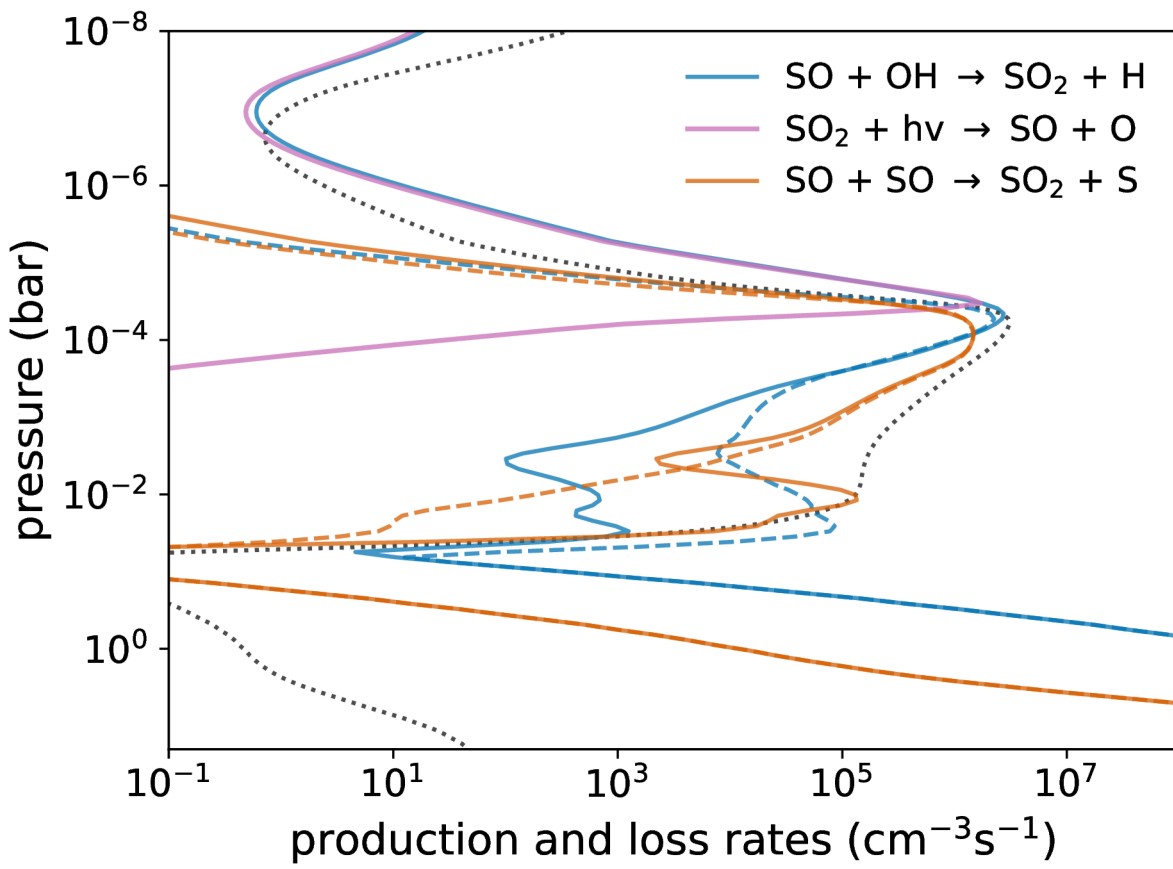

**Extended Data Fig. 4 | The main source and sink profiles of SO₂ in our WASP-39b model.** The reaction rates of the main sources and sinks of SO₂ in the VULCAN morning-terminator model for WASP-39b. The dashed lines of the same colour are the corresponding reverse reactions and the dotted black line indicates the distribution profile (arbitrarily scaled) of SO₂.

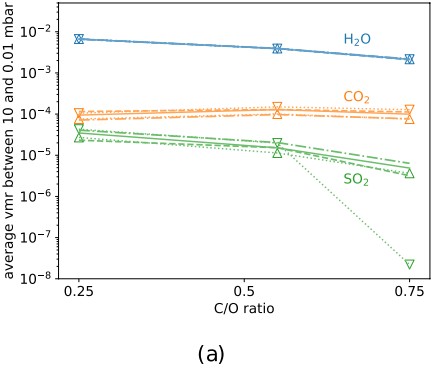

(a)

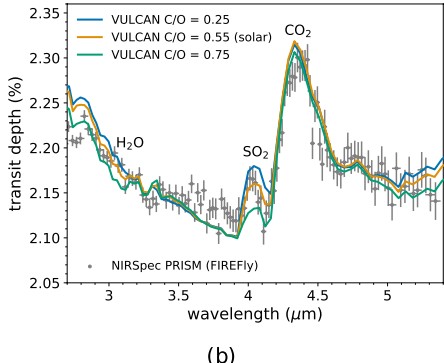

(b)

**Extended Data Fig. 5 | The C/O trends and synthetic spectra.** Same as Fig. 4 but as a function of C/O ratio at 10× solar metallicity. **a**, The averaged VMR of $H_2O$, $CO_2$ and $SO_2$ between 10 and 0.01 mbar as a function of C/O ratio, in which the solar C/O is 0.55. The nominal model is shown by solid lines, whereas the eddy diffusion coefficient ($K_{zz}$) scaled by 0.1 and 10 are shown by dashed and dashed-dotted lines, respectively. The models for which the whole temperature increased and decreased by 50 K are indicated by the upward-facing and downward-facing triangles connected by dotted lines, respectively. **b**, The morning and evening terminator-averaged theoretical transmission spectra with different C/O ratios compared with the NIRSpec PRISM observation. The error bars show $1\sigma$ standard deviations.

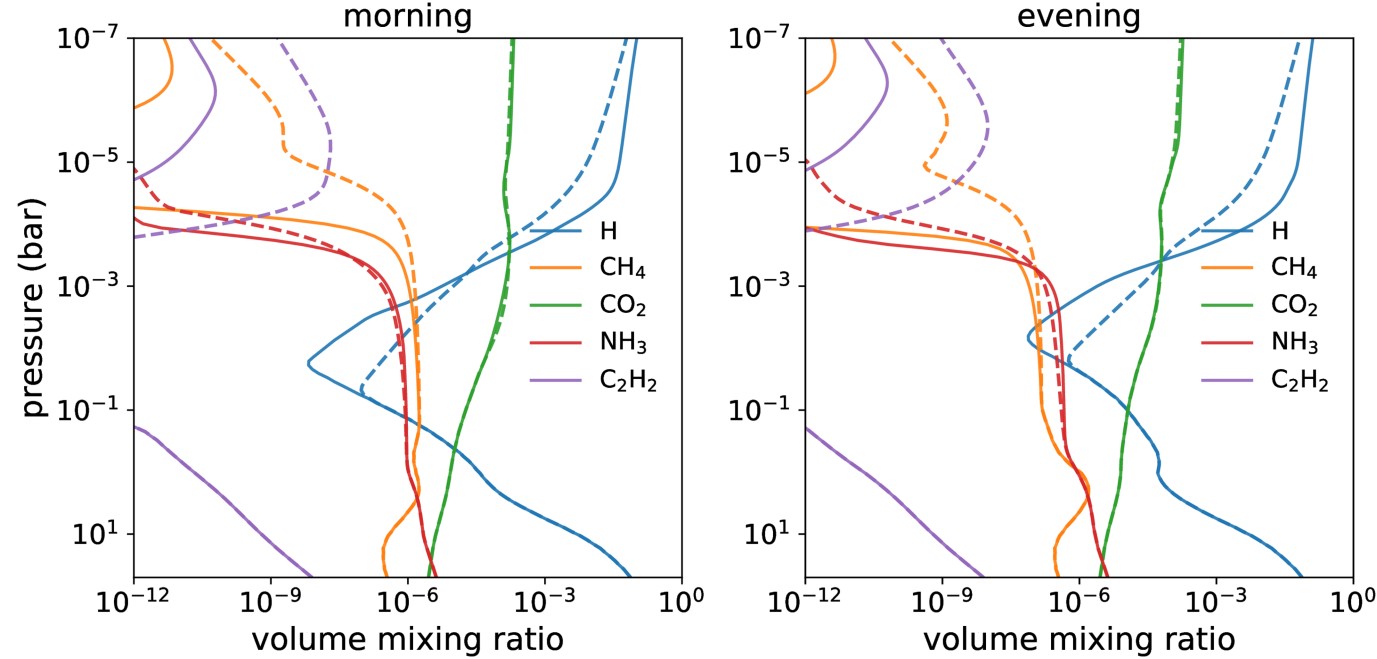

**Extended Data Fig. 6 | The impact of sulfur on other nonsulfur species.** VMR profiles of some species in our WASP-39b model that exhibit differences from VULCAN including sulfur kinetics (solid lines) and without sulfur kinetics (dashed lines).

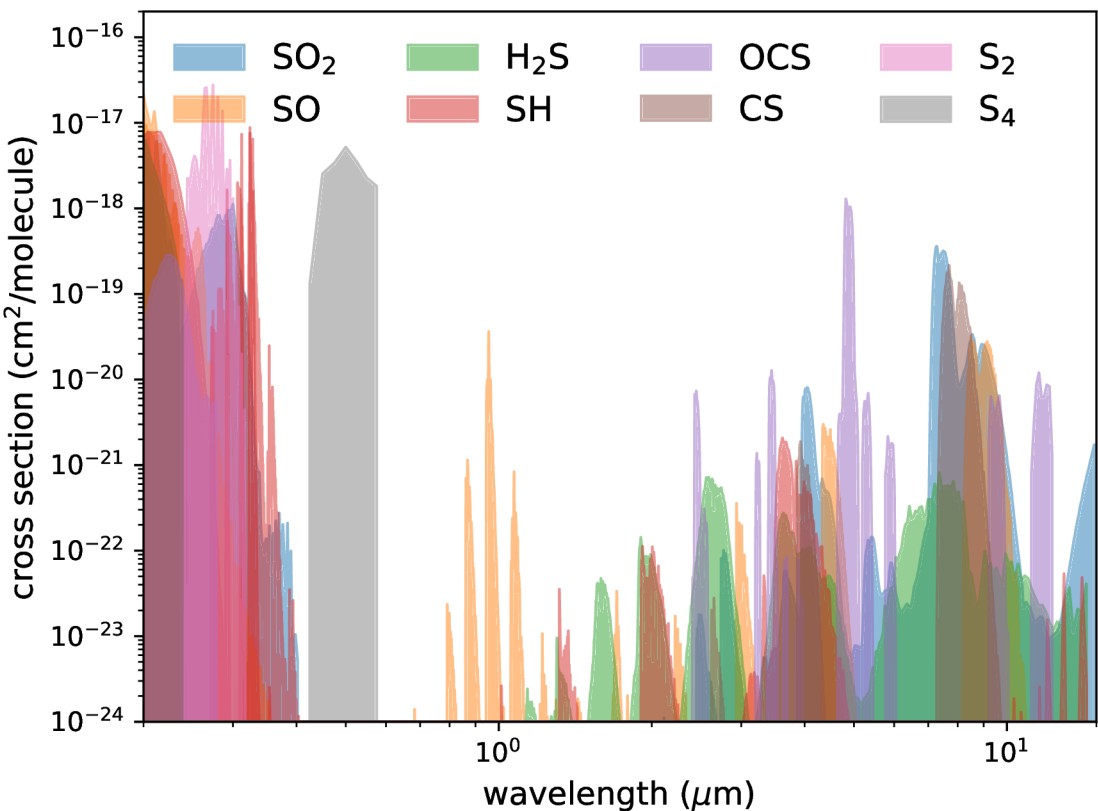

**Extended Data Fig. 7 | The opacities of several sulfur species.** Opacities of several sulfur species at 1,000 K and 1 mbar, except that those in the UV and of OCS are at room temperature. The opacities in the infrared are binned down to $R \approx 1,000$ for clarity.

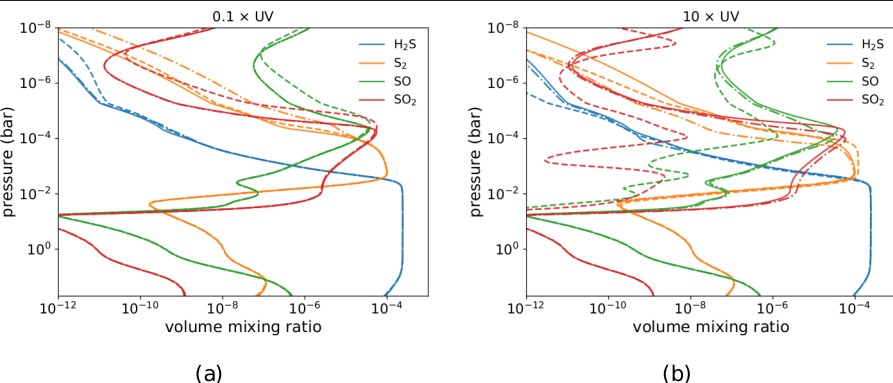

(a)                                                                    (b)

**Extended Data Fig. 8 | The main sulfur species abundances with reduced and enhanced UV irradiation.** VMR profiles of the main sulfur species in the VULCAN morning-terminator model with 0.1× (**a**) and 10× (**b**) UV. Our nominal model is shown by solid lines for comparison, whereas the model with varying FUV (1–230 nm) is shown by the dashed lines and that with varying NUV (230–295 nm) is shown by dashed-dotted lines.

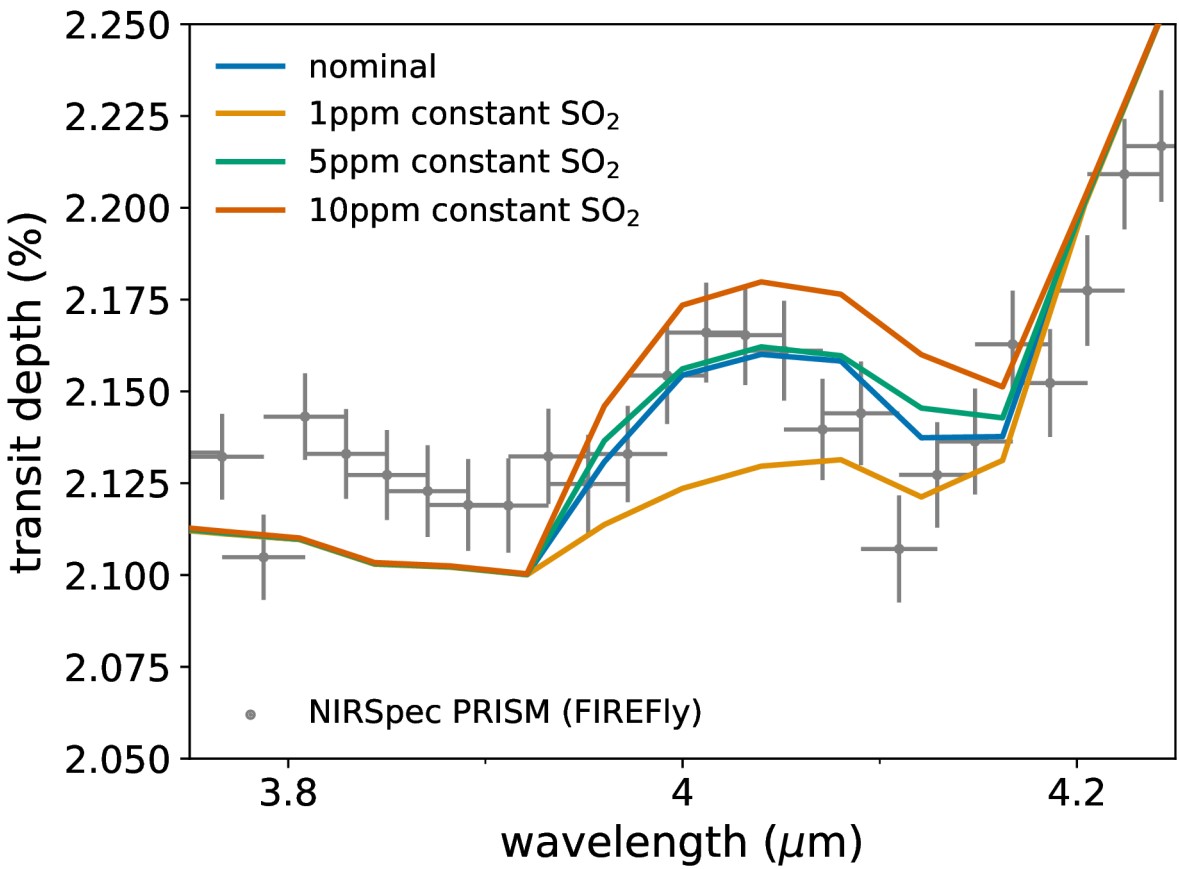

**Extended Data Fig. 9 | The effects of assuming a vertically uniform distribution of SO₂.** Terminator-averaged theoretical transmission spectra generated from abundance distribution computed by the photochemical model VULCAN compared with assuming constant 1, 5 and 10 ppm of SO₂. As before, the NIRSpec PRISM observation is shown with 1$\sigma$ error bars.

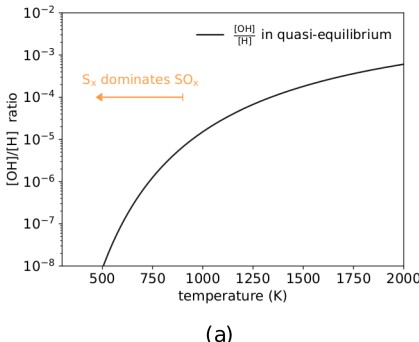
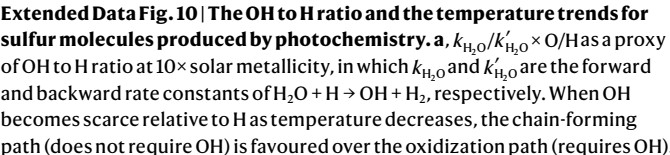

(a)

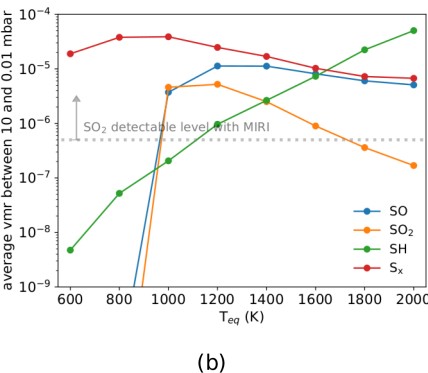

(b)

**Extended Data Fig. 10 | The OH to H ratio and the temperature trends for sulfur molecules produced by photochemistry. a**, $k_{H_2O}/k'_{H_2O} \times$ O/H as a proxy of OH to H ratio at 10× solar metallicity, in which $k_{H_2O}$ and $k'_{H_2O}$ are the forward and backward rate constants of $H_2O + H \rightarrow OH + H_2$, respectively. When OH becomes scarce relative to H as temperature decreases, the chain-forming path (does not require OH) is favoured over the oxidization path (requires OH).

**b**, The average VMR between 10 and 0.01 mbar as a function of planetary equilibrium temperature with temperature profiles adopted from ref. 39 (see text for the setup). The dotted grey line marks approximately the required $SO_2$ concentration to be detectable with WASP-39b parameters. $S_x$ denotes the allotropes $S_2$ and $S_8$ and $SO_x$ denotes the oxidized species SO and $SO_2$.

**Extended Data Table 1 | Important reactions for SO$_2$ production**

| Reaction | Rate Coefficient (cm$^3$ molecules$^{-1}$ s$^{-1}$) | Valid Temperature (K) | Ref. |
|---|---|---|---|
| H$_2$S + H $\longrightarrow$ SH + H$_2$ | 5.8 $\times 10^{-17}$ $T^{1.94}$ exp$(-455/T)$ | 190–2237 | [109] |
| SH + H $\longrightarrow$ S + H$_2$ | 2.16 $\times 10^{-11}$ | 295 | [133] |
| S + OH $\longrightarrow$ SO + H | 6.59 $\times 10^{-11}$ | 298 | [107] |
| SO + OH $\longrightarrow$ SO$_2$ + H | 1.79 $\times 10^{-7}$ $T^{-1.35}$ | 295–703 | [134] |
| S + SH $\longrightarrow$ S$_2$ + H | 4 $\times 10^{-11}$ | 295 | [135] |
| SO + SH $\longrightarrow$ S$_2$ + OH | 1 $\times 10^{-13}$ exp$(-2300/T)$ | | Est.[102] |
| SO + SO $\longrightarrow$ SO$_2$ + S | 3.5 $\times 10^{-15}$ | 298 | [136] |

List of selected reactions relevant for SO$_2$ production in the VULCAN model of WASP-39b.