## [Peer Review File · Nature]

Manuscript Title: Photochemically-produced SO₂ in the atmosphere of WASP-39b

Reviewer Comments & Author Rebuttals

Reviewer Reports on the Initial Version:

Referees' comments:

Referee #1 (Remarks to the Author):

The JWST observations of WASP-39b reported in Ahrer et al. 2022 revealed a transit feature around 4 μm that has motivated the investigation of sulfur chemistry for its origin. The current study provides a convincing interpretation of this observation as a signature of SO₂ formed by the atmospheric photochemistry.

The study benefits from the use of various independent models that arrive to similar conclusions regarding the abundance of the photochemically formed SO₂ under the same input of stellar flux, thermal structure and atmospheric mixing. As part of this evaluation another independent models was used with approximately the same input conditions and reached to similar conclusions. Therefore the presence of SO₂ at high enough abundance to reproduce the observed signature, under the assumed conditions, appears as a valid conclusion.

However there are various aspects in the proposed picture that require, at least, some discussion in the text. Once these are sufficiently addressed, the manuscript should be suitable for publication in Nature.

A. The first concern has to do with the considered temperature profiles presented in Extended Data Fig. 3. These profiles appear sufficiently cold to allow for the condensation of sulfur in various forms (MnS, Na₂S, etc.). The morning terminator, which presents the highest SO₂ abundance and possibly dominates the transit signature, appears much colder than the evening terminator and should certainly be affected by Na₂S condensation. However, both Na and sulfur should be present in the gas phase according to JWST observations. This contradiction deserves further discussion.

B. The second concern is again related to the assumed thermal structure and is related to the method of its evaluation. Although the use of a GCM is the optimal approach for evaluating the conditions at the two terminators, the thermal structure evaluation considers thermochemical equilibrium composition. The computational reasons for this approach are obvious, however the results of the study demonstrate that this approximation could be misleading. Figure 3 demonstrates that the opacity contributions from the sulfur photochemical products are significant, therefore could potentially affect the atmospheric thermal structure at a global level at the pressure region probed in the observations. 1D simulations of the thermal structure considering photochemical products at solar metallicity also suggest that sulphur components could have an impact on the thermal structure of hot-Jupiters (Lavvas & Arfaux, 2021). The authors do discuss a

sensitivity test of $\pm 50\text{K}$ but it is not clear if this temperature uncertainty is sufficient for the anticipated impact of the sulfur photochemical products, given the high abundance implied by the observed transit depth. In view of the first concern, this is an aspect that should be considered by the authors.

C. The abstract argues that sulphur photochemistry could be observable at equilibrium temperatures above $\sim 750\text{ K}$. This argument is based on previous studies that identified a significant production of sulphur allotropes at lower temperatures. However, there is no quantitative evaluation in this work about the relative contributions of the two components at different temperatures. Moreover, as mentioned also in the manuscript, at higher temperatures than those assumed, the SO_2 abundance is decreased in favor of SO and SH . These molecules would be harder to identify in the transit spectra, therefore the temperature range where sulphur photochemistry could be well characterized is rather bracketed by these two mechanisms.

Some further minor comments are provided below.

p.164-167: - Are there other candidates that could be participating in the observed feature?

- Why the presence or abundance of SO_2 was not supported?

Due to lower metallicity assumptions or lack of sufficient reaction network? Also this sentence appears in disagreement with the sentence in lines 185-186.

Fig. 2: The “cyan” arrows appear to be green.

Fig. 3: Please add the corresponding transit spectrum without SO_2 for one of the models, to demonstrate the strong features anticipated in the MIRI LRS wavelength range.

p. 272-274: The conclusion about SO_3 is not substantiated by any arguments or other studies.

p. 303-305: In general the manuscript lacks a description of the source of the most important rates considered in the modeling. These could perhaps be different among the different models, but there are not too many options available for the most important reactions identified. Even though a link to the whole reaction network of VULCAN is provided, a quick discussion would be beneficial.

Moreover, it would be good to be more explicit about which rate constants and UV cross sections need to be better evaluated for the SO_2 photochemistry.

Extended Data Fig.3: It would be good to present the estimated eddy profile based on the GCM results along with applied profile from eq. 2.

p.1179: Missing units for column density.

Referee #2 (Remarks to the Author):

1: Summary of key results

1D photochemical models robustly reproduce the presumed SO₂ absorption feature seen in transit observations of WASP-39b. The strength of the SO₂ absorption feature is highly sensitive to atmospheric metallicity, and WASP-39b looks to have ~10x solar metallicity. Given the best fit models to the observations considered in this study, the authors make predictions of additional SO₂ absorption features observable at shorter and longer wavelengths, and of additional CO₂ absorption features at longer wavelengths. When future studies incorporate SO₂ and other photochemically produced species into retrieval frameworks, they should be wary of parameterizing with uniform atmospheric abundances. This is particularly exciting because measurements of SO₂ will likely be useful in efforts to link atmospheric compositions to planet formation scenarios.

2: Originality and significance >> “does this paper represent an advance in understanding likely to influence thinking in the field, with strong evidence for their conclusions.”

Conclusions are original, to the best of my ability to search through the literature.

I think this paper will influence the field of exoplanet studies to pay attention to photochemistry (and in particular sulfur photochemistry) when interpreting observations of hot jupiters... and to consider models which extend upwards of 10x solar more often. They include a nice section in the extra data which should influence the field’s approach to atmospheric retrievals on exoplanet data by showing that one cannot treat SO₂ as vertically uniform in distribution throughout the atmosphere, a conclusion which likely generalizes to other gasses affected by photochemistry. The ability to assess sulfur abundances could also help with efforts to link atmospheric measurements to formation histories, which is a major goal for many in the field. The authors also make predictions of additional SO₂ and CO₂ features at other wavelengths. It looks like there is one other paper that has discussed the likelihood of detecting SO₂ in JWST transit spectra that was just accepted for publication right around when I assume this study was wrapping up/being submitted. The two works are complementary since one is considering a broader swath of parameter space rather than focusing on WASP 39-b specifically.

Detecting SO₂ as a product of photochemistry is among the first of hopefully many new exoplanet atmosphere-related “surprises” to be found by JWST. Published literature going back years had looked at sulfur photochemistry, including SO₂. But it seems no one had noted its detectability at this wavelength until extremely recently. Presumably the reason SO₂ detection had not been anticipated much earlier is that prior studies considering implications for transit spectra had not run detailed hot jupiter-condition sulfur photochemical models at high enough metallicities.

Overall, I think the authors make a very convincing case that photochemistry naturally produces the observed absorption feature via SO₂.

3: Data & methodology: validity of approach, quality of data, quality of presentation

The authors utilized an array of peer reviewed modeling tools appropriate to the task they had set

themselves. JWST and HST are state of the art observing facilities for collecting transit spectra. Presentation of methods and data are concise but clear.

4: Appropriate use of statistics and treatment of uncertainties

None of the error bars are defined in the figures which include spectral data.

Rather than statistical tests, the authors conduct sensitivity tests which are all done by varying model assumptions within “reasonable” ranges and demonstrating that the conclusion that photochemistry can produce the right amount of SO₂ to replicate the transit spectrum remains true. These tests seem appropriate to me, and quite comprehensive. I will list them below:

Photochemistry: Used 4 independently developed 1D photochemistry models. In each case choices must be made in selecting chemical reaction network, kinetics data, and/or numerical solver. All are still 1D models though, so there was no comparison to 2D or 3D models which can capture horizontal transport.

P-T profiles: taken from a GCM output up to 5×10^{-5} bars, then set to isothermal. Tested +/- 50 K for the sensitivity study

Internal heat: 358 K, state that varying this value didn't change abundances of observable species at all

Metallicity: 5, 10 and 20x solar

% ratio: 0.25, 0.55, 0.75

UV irradiation: Compared fiducial adopted spectrum to results with the solar spectrum, and also varied UV by factor of 10, then NUV and FUV separately by factors of 10. Found that the only change came when FUV was increased by a factor of 10.

Kzz: approximated from GCM. Multiplied and divided by a factor of 10 for sensitivity study.

5: Conclusions: robustness, validity, reliability

Aside from the requested changes below, I am satisfied that this work's conclusions are robust, valid, and reliable.

6: Suggested improvements, experiments, data for possible revision

With a small number of changes, I think this manuscript should be published! I have made a chronological list of suggested changes, comments, and questions for the authors.

I have one general comment about the paper overall: It seems like it is missing a statement along the lines of, “there is no other plausible scenario to produce this spectral feature”. Or “xyz is an

alternative scenario, but we rule it out because...” Can you make any statements along those lines? Even if this is obviously the correct explanation to experts, it will be helpful for readers who are not as familiar with this subject. I’m quite convinced by your work, but it seems like an explicit acknowledgement of alternative possibilities or lack-there-of would round out the logic of your assertion that this is “direct evidence of photochemistry”. If not, it seems like you should shift the headline message of the paper away from “this is beyond a doubt the first direct signature of photochemistry in an exoplanet” and towards “photochemistry can robustly reproduce an SO₂ absorption signature which agrees with this data”.

Line 164-167: I find this sentence misleading. It doesn’t seem to me that SO₂’s presence in the WASP-39b transit spectrum was “not yet supported” by the existing modeling tools. This paper’s main conclusions did not require the introduction of additional physical or chemical processes into modeling tools, rather it required the application of existing tools at the appropriate place in temperature-metallicity-% ratio parameter space. I think that moving the concluding sentence of the following paragraph to the beginning of said paragraph and rephrasing a bit would help clarify this point. It also seems appropriate to mention Polman 2022 around here somewhere— although they mostly emphasize the longer-wavelength feature at ~7 microns, if one looks at their Fig. 6 the ~4 micron feature can be seen in the 10x metallicity, low % ratio case.

Line 189: You state that you perform cloud-free photochemical model calculations. Best fit models in Rustamkulov et al submitted and Alderson et al submitted included clouds. To what extent would including this additional opacity source affect your results? Will it influence UV penetration into the atmosphere enough to change SO₂ distribution? Or is photochemistry happening above the clouds?

Same paragraph starting at Line 189: Later in the conclusions around lines 301-306 you finally mention 2D chemical models and considerations of horizontal transport. Prior to that you imply that these different 1D models encapsulate the full range of model uncertainty. Are the expected changes in chemical abundances due to horizontal transport on the same order as differences from “reasonable” variation in 1D models’ chemical networks, kinetics data, and numerical design? Since you have run a GCM already it seems like some horizontal wind speeds near the terminators could be utilized for a mixing vs chemical timescale comparison without too much additional work needed on your part.

Line 220-223: Do S, S₂ and SO produce detectable features outside of the PRISM/G395H wavelength range?

Main article Figure 3 and discussion of it (Lines 240-257): Please include a panel with the 1-3 micron wavelength coverage. This way readers can assess agreement across the full wavelength range, not just the portion surrounding the SO₂ feature. Around 3.5-4 microns the models look like they have shallower transit depths than the data, whereas in Rustamkulov et al, submitted and Alderson et al submitted this does not appear to be the case. Can you comment on why this is? Do you include clouds in your spectral models? In the caption, be sure to note the method of calculating the error bars shown in the figure, per Nature’s policy.

Line 254: I would state more explicitly here that the UV opacities are only valid at room

temperature, if that is what is meant by saying the discrepancy “could be potentially due to enhanced UV opacities at high temperatures”.

Lines 288-289: I think your references here ought to mention Polman 2022 again since it is the one other study that noted the observability of SO₂ in this temperature-metallicity-% regime.

Lines 1131-1145: If there is room and the lines don't simply lie right on top of each other, I think it would be informative to include panels for other sensitivity tests besides just % ratio in extended data figure 5. If you are needing to limit the number of figures/make space, I did not think that Extended Data Fig. 1 enhanced my understanding much. The stated numbers in lines 176-178 conveyed the point on their own. However if you left them out because there's not a visible trend anyway, then leaving things as is seems like a fine choice.

Extended Fig. 7: I'm not sure if it was just an issue with my version of adobe reader, but this figure caused scrolling to be laggy.

7: References

I think appropriate credit was given to previous work, aside from a few of the suggested re-phrasings above. I did not come across any missing references.

8: Clarity and context

The writing is very clear and easy to follow. The experimental design and modeling choices are well justified and adequately explained. Aside from my suggestions above, I am very satisfied on this front.

Author Rebuttals to Initial Comments:

We greatly thank the referees' remarks and critical comments. Here are our responses to the major comments of Referee #1

A. The temperature profiles of WASP-39b indeed cross several condensation curves, including sulfide clouds. Nevertheless, the gaseous compositions are not expected to be significantly affected by condensation based on the following two reasons. The high surface energies of the sulfide condensates (e.g., Gao. et al. 2020, Yu et al. 2021) suggest that their nucleation rates are low and unlikely to form abundant particles. As the Referee pointed out, the presence of both Na and sulfur in the gas phase from the JWST's spectra supports this argument. Secondly, Na is only about 10% of S based on the solar elemental ratio. Hence even in the extreme case, the full condensation of Na₂S can only deplete about 20% of sulfur at most. We have added the discussion in the first section of Methods.

Gao. et al. 2020

<https://ui.adsabs.harvard.edu/abs/2020NatAs...4..951G/abstract>

Yu et al. 2021

<https://ui.adsabs.harvard.edu/abs/2021NatAs...5..822Y/abstract>

B.

To evaluate the radiative feedback from the disequilibrium chemical abundances, we have performed a 1D self-consistent radiative-transfer-kinetics model using HELIOS--VULCAN, where H₂S, SH, and SO₂ are included in HELIOS for the opacity sources of sulfur species.

Yet we found negligible differences between the temperature profile computed from equilibrium abundances and that from disequilibrium abundances. This is likely because water, which dominates the infrared opacity source, remains unaltered by photochemistry and vertical mixing.

However, parts of the opacities are missing in our radiative-transfer calculations (main sources: ExoMol and HITEMP/HITRAN; see references in Malik et al. 2019). In particular, the opacity of SO₂ (Gorman et al. 2019) does not extend to the UV and S₂ is not included. As pointed out by the Referee and Zahnle et al. (2019); Lavvas & Arfaux (2021), SH and S₂ can potentially impact the thermal structure with their strong absorption in the UV—visible wavelength range. We have now added a discussion “Radiative feedback of disequilibrium composition” in Methods with Extended Data Fig. 4 to further quantify the radiative heating by these sulfur photochemical products. Our estimate shows that SO₂ contributed the most to shortwave heating on WASP-39b, rather than SH and S₂ being the main shortwave absorbers for solar-like metallicity atmospheres in Zahnle et al. (2019) and Lavvas & Arfaux (2021). The heating due to SO₂ is comparable to a grey opacity of 0.05 and could potentially raise the temperatures around 0.1 mbar. Nevertheless, the heating effect does not change our main conclusion in this work about interpreting SO₂ formation on WASP-39b. As long as the temperature does not fall below ~750 K where S_x formation starts to take over, the SO₂ distribution is not too sensitive to temperature increase up to ~100 K.

C.

We agree with the Referee that at high temperatures, the main molecules in the observable part of the atmosphere, SH and SO, are more challenging to identify. Yet it is feasible to identify SH in the NUV—optical spectra (e.g. Mikal-Evans 2019). We have added the section “Implications of observing sulphur photochemistry” in Methods to discuss the mechanism governing the contribution of sulfur allotrope and sulfur oxides and to provide a quantitative evaluation of the temperature trends for sulfur photochemical products.

For specific minor comments:

p.164-167: - Are there other candidates that could be participating in the observed feature?

An extensive list of species has been considered (Rustamkulov et al.). In particular, H₂S, HCN, HBr, PH₃, SiO, and SiO₂ have features close to 4 micron but were ruled out by the precision of JWST data.

H₂S and HCN absorb shortward to the feature at 4.05 micron, whereas HBr, PH₃, and SiO have a little wider absorption than the observed feature.

- Why the presence or abundance of SO₂ was not supported?

Due to lower metallicity assumptions or lack of sufficient reaction network? Also this sentence appears in disagreement with the sentence in lines 185-186.

We meant to say that the recent ERS analysis of Wasp-39b (Rustamkulov et al., Alderson et al.) using injected uniform SO₂ to explain the observed spectra were not supported by

physical models. We have revised the sentences to avoid confusion.

Fig. 2: The “cyan” arrows appear to be green.

They are actually from the cyan color palette. We have changed the description to dark cyan.

Fig. 3: Please add the corresponding transit spectrum without SO₂ for one of the models, to demonstrate the strong features anticipated in the MIRI LRS wavelength range.

Done.

p. 272-274: The conclusion about SO₃ is not substantiated by any arguments or other studies.

This is based on the photochemical models used in this study. None of the models produced SO₃ with abundances higher than 10⁻¹⁰ VMR.

p. 303-305: In general the manuscript lacks a description of the source of the most important rates considered in the modeling. These could perhaps be different among the different models, but there are

not too many options available for the most important reactions identified. Even though a link to the whole reaction network of VULCAN is provided, a quick discussion would be beneficial. Moreover, it would be good to be more explicit about which rate constants and UV cross sections need to be better evaluated for the SO₂ photochemistry.

We have identified the main pathways for producing SO₂ (1), where all of the reactions involved in the pathway have rate consistent coefficients on the NIST database. The main sources and sinks of SO₂ are also illustrated in Extended Data Fig. 4. We have added Extended Data Table 1 to specify the important rate constants used in VULCAN and the full network used by other models are also available on XXX.

Extended Data Fig.3: It would be good to present the estimated eddy profile based on the GCM results along with applied profile from eq. 2.

Done.

p.1179: Missing units for column density.

Added.

Here are our responses to Referee #2

It seems like it is missing a statement along the lines of, “there is no other plausible scenario to produce this spectral feature”. Or “xyz is an alternative scenario, but we rule it out because...” Can you make any statements along those lines? Even if this is obviously the correct explanation to experts, it will be helpful for readers who are not as familiar with this subject. I’m quite convinced by your work, but it seems like an explicit acknowledgement of alternative possibilities or lack-there-of would round out the logic of your assertion that this is “direct evidence of photochemistry”. If not, it seems like you should shift the headline message of the paper away from “this is beyond a doubt the first direct signature of photochemistry in an exoplanet” and towards “photochemistry can robustly reproduce an SO₂ absorption signature which agrees with this data”.

We have added some sentences stating other plausible scenarios we have ruled out at the end of the first paragraph to better convey the logical progression.

Line 164-167: I find this sentence misleading. It doesn’t seem to me that SO₂’s presence in the WASP-39b transit spectrum was “not yet supported” by the existing modeling tools. This paper’s main conclusions did not require the introduction of additional physical or chemical processes into modeling tools, rather it required the application of existing tools at the appropriate place in temperature-metallicity-C/O ratio parameter space. I think that moving the concluding sentence of the following paragraph to the beginning of said paragraph and rephrasing a bit would help clarify this point. It also seems appropriate to mention Polman 2022 around here somewhere—although they mostly emphasize the longer-wavelength feature at ~7 microns, if one looks at their Fig. 6 the ~4 micron feature can be seen in the 10x metallicity, low C/O ratio case.

As also pointed out by Referee #2, we have revised the sentence to “although ad-hoc spectra with injected SO₂ were used in the analysis.” We have also included Polman et al. (2022) in the following paragraph.

Line 189: You state that you perform cloud-free photochemical model calculations. Best fit models in Rustamkulov et al submitted and Alderson et al submitted included clouds. To what extent would including this additional opacity source affect your results? Will it influence UV penetration into the atmosphere enough to change SO₂ distribution? Or is photochemistry happening above the clouds?

In terms of interpreting the observed transmission spectra, clouds damped the spectral features. Clouds are treated as a grey opaque layer with varying cloud deck pressures in the best-fit models in Rustamkulov et al. and Alderson et al.

We did also find that including grey clouds improves our model fit, especially the broad water band (2.8–3.6 micron). We chose to present cloud-free photochemical models without adding clouds for direct comparison.

In terms of whether the existence of clouds would affect the results of photochemical models by shielding the stellar UV flux, it depends on the altitude of clouds.

Photodissociation in an H₂ atmosphere is dominated by FUV up to about 200 nm, which does not penetrate deep but is absorbed around micro-bar levels (e.g., Fig. 17 in Tsai et al. 2021).

The clouds in the best-fit models in Rustamkulov et al and Alderson et al are generally much lower than the photolysis level (cloud top at 0.3 mbar in PHOENIX and around a few mbar in PICASO-VIRGA).

Hence we do not expect the low clouds to significantly affect the photochemical processes and chemical distributions, unless thick, high-altitude hazes are present. Additionally, Extended Data Fig. 8 shows that the results are not sensitive to the case even when the UV is obscured down to 10% since it is not in the photon limited regime.

Same paragraph starting at Line 189: Later in the conclusions around lines 301-306 you finally mention 2D chemical models and considerations of horizontal transport. Prior to that you imply that these different 1D models encapsulate the full range of model uncertainty. Are the expected changes in chemical abundances due to horizontal transport on the same order as differences from “reasonable” variation in 1D models’ chemical networks, kinetics data, and numerical design? Since you have run a GCM already it seems like some horizontal wind speeds near the terminators could be utilized for a mixing vs chemical timescale comparison without too much additional work needed on your part.

The fast horizontal winds in our GCM, around 0.5 — 2 km/s above 0.1 bar, suggest a horizontal transport timescale in the order of a few days. This is much shorter than the typical thermochemical timescales (e.g., Tsai et al. 2018) and vertical mixing timescale, suggesting that zonal wind should efficiently homogenize the compositional variations in the horizontal direction omitting photochemistry. However, photochemistry usually has a shorter or comparable timescale to the

horizontal transport timescale here and numerical models are required to quantify how much global variations to expect.

For example, Agundez et al. (2014) used a pseudo 2D model (rotating 1D column) to show that for a hotter planet, HD 189733b, neglecting zonal transport can cause the abundance of some species differ about an order of magnitude, on the same order as the uncertainties of our suite of 1D photochemical models.

However, with a similar pseudo 2D model, Venot et al. (2020) indicates that for WASP-43b, neglecting zonal transport can change the abundances of certain species more than two orders of magnitude when vertical mixing is very efficient. Nevertheless, we will leave a thorough analysis incorporating sulfur kinetics in 2D/3D models for future work.

%%Baeyens et al. (2022) show that for an atmosphere with similar temperature and host star to WASP-39b, the horizontal variations

Line 220-223: Do S, S₂ and SO produce detectable features outside of the PRISM/G395H wavelength range?

Yes, we discussed how S₂ along with SH and SO₂ could be observed in the UV in the following paragraph after next. Atomic S has strong lines around 140 and 180 nm and has been observed on Io by Hubble/STIS.

Detecting SO might be more challenging due to its weaker lines in the IR and overlapping absorption with SH in the EUV.

Main article Figure 3 and discussion of it (Lines 240-257): Please include a panel with the 1-3 micron wavelength coverage. This way readers can assess agreement across the full wavelength range, not just the portion surrounding the SO₂ feature. Around 3.5-4 microns the models look like they have shallower transit depths than the data, whereas in Rustamkulov et al, submitted and Alderson et al submitted this does not appear to be the case. Can you comment on why this is? Do you include clouds in your spectral models? In the caption, be sure to note the method of calculating the error bars shown in the figure, *per Nature's policy*.

We did not include clouds in our spectral models, since we intended to focus on the comparisons with cloud-free photochemical models.

We have extended the wavelength to 1.5 micron in first panel but still incline to focus on the spectra are impacted by photochemistry.

We defer the readers to other ERS Wasp39b papers where equilibrium models over a reasonably broad range parameter space are used to fit the spectra, especially the water features with various metallicity, C/O, and clouds.

The shallower part of the spectra around 3.5-4 microns where no strong gas opacity is present in this wavelength range is likely due to clouds obscuring the lower atmosphere. We have added the definition of error bars in the figure legend.

Line 254: I would state more explicitly here that the UV opacities are only valid at room

temperature, if that is what is meant by saying the discrepancy “could be potentially due to enhanced UV opacities at high temperatures”.

We have added the description.

Lines 288-289: I think your references here ought to mention Polman 2022 again since it is the one other study that noted the observability of SO₂ in this temperature-metallicity-% regime.

We have added the reference.

Lines 1131-1145: If there is room and the lines don't simply lie right on top of each other, I think it would be informative to include panels for other sensitivity tests besides just % ratio in extended data figure 5. If you are needing to limit the number of figures/ make space, I did not think that Extended Data Fig. 1 enhanced my understanding much. The stated numbers in lines 176-178 conveyed the point on their own. However if you left them out because there's not a visible trend anyway, then leaving things as is seems like a fine choice.

We have explored the dependence of or sensitive to metallicity, C/O ratio, vertical mixing (K_{zz}), temperature, and stellar UV fluxes, as shown in Fig. 4 Extended Data Fig. 5 and 8. The most desirable trend in metallicity is discussed in the main text.

We consider this parameter space to be sufficient for this study focusing on WASP-39b and will defer studies with a broader parameter space to Polman et al. (2022) and future studies.

Reviewer Reports on the First Revision:

Referees' comments:

Referee #1 (Remarks to the Author):

The authors have addressed clearly and sufficiently the previous comments. I appreciate their time and effort to address the concern for the implications of disequilibrium on the thermal structure. The manuscript will be ready and suitable for publication once the following minor points are corrected.

In the "Radiative Feedback of Disequilibrium Composition" section please use the same symbols for the equation and the following discussion for the δ_{mi} or δ_{mi} etc. It would also be good to present the simulated disequilibrium p-T profile.

In the "Implications of Observing Sulphur Photochemistry" section, last paragraph: "For $T_{eq} \lesssim 1000$ K, SO₂ production below the 0.01 mbar level IS ceases and Sx...". Probably you don't need the "IS" in the sentence.

In Extended Data Fig. 11:

- The parentheses in the middle of the paragraph with the (required/not required OH) seem to be in contradiction with what the message should be. Please review and correct.
- SO_x does not appear in the figure caption, SO and SO₂ have their individual lines, so the last sentence of the caption needs to be modified.

These modifications can be reviewed by the editor, I do not have to see the updated manuscript.

[REDACTED]

Referee #2 (Remarks to the Author):

I am satisfied that this manuscript is ready for publication.

Author Rebuttals to First Revision:

We have revised all the comments by Reviewer #1. Except regarding the suggestion "*It would also be good to present the simulated disequilibrium p-T profile.*"

As we pointed out in the text, the disequilibrium p-T profile appears almost identical to the equilibrium p-T profile when the shortwave opacity of SO₂ is not considered thus not worth presenting it.

Regarding "*SO_x does not appear in the figure caption, SO and SO₂ have their individual lines, so the last sentence of the caption needs to be modified.*"

SO_x actually does appear in the left panel.